# Robust Equilibria in Continuous Games: From Strategic to Dynamic Robustness

**Kyriakos Lotidis**
Stanford University
klotidis@stanford.edu

**Panayotis Mertikopoulos**
Univ. Grenoble Alpes, CNRS, Inria, Grenoble INP
LIG 38000 Grenoble, France
panayotis.mertikopoulos@imag.fr

**Nicholas Bambos**
Stanford University
bambos@stanford.edu

**Jose Blanchet**
Stanford University
jose.blanchet@stanford.edu

## Abstract

In this paper, we examine the robustness of Nash equilibria in continuous games, under both strategic and dynamic uncertainty. Starting with the former, we introduce the notion of a *robust equilibrium* as those equilibria that remain invariant to small—but otherwise arbitrary—perturbations to the game's payoff structure, and we provide a crisp geometric characterization thereof. Subsequently, we turn to the question of dynamic robustness, and we examine which equilibria may arise as stable limit points of the dynamics of *"follow the regularized leader"* (FTRL) in the presence of randomness and uncertainty. Despite their very distinct origins, we establish a structural correspondence between these two notions of robustness: strategic robustness implies dynamic robustness, and, conversely, the requirement of strategic robustness cannot be relaxed if dynamic robustness is to be maintained. Finally, we examine the rate of convergence to robust equilibria as a function of the underlying regularizer, and we show that entropically regularized learning converges at a geometric rate in games with affinely constrained action spaces.

## 1 Introduction

A fundamental requirement in game theory—which predates even the cornerstone notion of a Nash equilibrium—concerns the robustness that should be inherent in any axiomatization of rational behavior. To quote a famous passage by von Neumann & Morgenstern [53, p. 32]: "In whatever way we formulate the guiding principles and the objective justification of rational behavior, provisos will have to be made for every possible conduct of "the others." If the superiority of rational behavior over any other kind is to be established, then its description must include rules of conduct for all conceivable situations—including those where "the others" behaved irrationally in the sense of the standards which the theory will set for them."

As a byproduct of this tenet, there has been a flurry of activity since the 1970s in proposing refinements of the Nash equilibrium concept, all in an effort to dismiss equilibria that are highly fragile or otherwise implausible (e.g., because they involve threats that are not credible).[1] This pursuit of robustness has recently gained increased momentum owing to the applications of game theory to machine learning and data science, two fields where the notion of robustness has been likewise elusive. Here, even though many game-theoretic solutions perform extremely well on specific tasks—such as a well-trained generative adversarial network (GAN) at equilibrium—the resulting models tend to

---

[1]For a masterful introduction to the topic, see the textbook of van Damme [52].

39th Conference on Neural Information Processing Systems (NeurIPS 2025).

have a narrow performance envelope, being brittle, and unable to adapt to situations that deviate from their initial configuration.

In game-theoretic terms, this highlights the fact that, even though a Nash equilibrium is resilient to unilateral deviations, it need not be *robust* to small perturbations in the *payoff data* of the game (which, in a machine learning context, could represent distributional shifts, incomplete observations, and/or other sources of uncertainty). In view of this, it is natural to ask

*Which equilibria remain robust in the presence of strategic uncertainty?*

This question has been the lodestar of the equilibrium refinement literature, and it has led to a wide array of proposals aiming to get rid of "unreasonable" equilibria that may disappear even under the most minute perturbation to the players' payoffs—from Selten's notion of *trembling hand perfection* [46], to Myerson's concept of *properness* [38], and the various criteria of strategic stability introduced by Kohlberg & Mertens [28] (hyperstability, full stability, sequential stability, etc.).

Dually to the above theory of "strategic refinement", an important alternative approach has been based on *dynamic* considerations: that is, the players of a game start off-equilibrium, and in one sense or another learn (or fail to learn) to play an equilibrium over time. Here, the focus is on the players' learning protocol, the information available during play, and the presence (or absence) of players that may deviate from this protocol. By the so-called "folk theorem of evolutionary game theory" [23], it is well known that only *strict* equilibria are stable and attracting under the replicator dynamics, a result which was extended more recently to a broad class of "regularized learning" schemes, in both continuous [17] and discrete time [18, 19, 36].

These two viewpoints are not always compatible: for instance, in $2 \times 2$ games with two pure equilibria and one mixed (such as the Chicken / Hawk-Dove game), the mixed equilibrium is ruled out by almost all game-theoretic learning algorithms and dynamics, even though it survives a broad range of strategic refinement attacks. A point of hope here is the equivalence between (setwise) strategic and dynamic stability proved by Ritzberger & Weibull [40], who showed that a span of pure strategies in the mixed extension of a finite game is strategically stable in the sense of Kohlberg & Mertens [28] if and only if it is asymptotically stable under the replicator dynamics—see also [10, 13] for an extension to a wider class of discrete-time models for learning, with different information assumptions.

Notably, these considerations all concern finite games in normal (or extensive) form. By contrast, most applications of game theory to machine learning and data science involve *continuous games*, that is, games with a finite number of players and a continuum of actions per player—for example, GANs, multi-agent reinforcement learning, Kelly auctions, etc. In view of this, our paper seeks to answer the following questions in the context of continuous games:

*Which equilibria arise as robust predictions of the players' learning dynamics?*

We refer to these two types of robustness as *strategic* and *dynamic robustness*, respectively. Our paper seeks to quantify the interplay between the two, and the implications that connect them.

**Our contributions in the context of related work.**     Aiming for the strongest possible definition of robustness, we propose the following strategic refinement criterion:

*An equilibrium of a continuous game is strategically robust*
*if it remains an equilibrium in any slightly perturbed, nearby game.*

This requirement is similar in spirit to—but considerably stronger than—the classical notion of *essentiality* of Wu & Jiang [55], which posits that any nearby game has a nearby, possibly different equilibrium. Importantly, our results apply to *local* Nash equilibria, which are especially relevant in machine learning applications where payoff landscapes are typically nonconcave. This distinction is crucial, as global Nash equilibria do not always exist in general continuous games, making local equilibrium guarantees both meaningful and necessary in practice.

An important point here is that, in contrast to finite games—where the notion of "nearby" is fairly unambiguous—perturbations to a continuous game involve functional variations and, as such, the metric that quantifies a "small" perturbation plays a crucial role. Importantly, albeit natural, our proposed robustness requirement becomes vacuous if distances are measured with respect to the

players' payoff functions: more precisely, it is always possible to find a payoff perturbation with arbitrarily small $L^\infty$-norm that ends up upsetting *any* equilibrium.

The underlying issue here is that a small payoff perturbation may exhibit very high local variability, which can disrupt the first-order stationarity conditions that characterize equilibria in continuous games, thereby eliminating them altogether. To circumvent this issue, we argue that deviations of continuous games should be measured by comparing their respective *gradient fields*, which encode all the strategic information in the game. This shift in perspective leads to a crisp geometric characterization of strategically robust equilibria: they are extreme points of the game's action space, and they are sharp in the sense that the game's individual payoff gradients form a strictly acute angle with any tangent direction (cf. Fig. 1 later in the paper).

From a dynamic standpoint, we focus throughout on the family of algorithms known as *"follow the regularized leader"* (FTRL) [31, 47–49]. This is arguably one of the most—if not *the* most—popular class of policies for online learning due to its strong regret minimization and convergence guarantees, and it contains as special cases gradient descent/ascent methods [3, 58], dual averaging [39, 56], the exponential / multiplicative weights algorithm [2, 5, 5, 33, 54], implicitly normalized forecasters [1, 4, 57], exponentiated gradient methods [7, 27, 51], and many stochastic approximation schemes, adaptive [24, 25] and non-adaptive alike [26, 34–36].

In this general context, we examine which equilibria admit robust convergence guarantees as stable limit points of the dynamics of FTRL in the presence of randomness and uncertainty. Our first main result is that *strategic robustness implies dynamic robustness*, i.e., any strategically robust equilibrium is stable and attracting with high probability under the dynamics of FTRL, for any choice of regularizer. Conversely, we also show that the strategic robustness requirement cannot be lifted, and we provide an example of a game with an extreme, non-robust equilibrium which attracts *all* FTRL orbits under a certain choice of regularizer, and *none* under another.

To the best of our knowledge, this is the first result of its kind for general continuous games. In the context of *finite* games, Flokas et al. [17] showed that a point is asymptotically stable under the continuous-time FTRL dynamics if and only if it is a strict Nash equilibrium, while [10] extended this equivalence to discrete-time models of regularized learning under uncertainty. Strict equilibria are prime examples of strategically robust equilibria, so this part of the analysis of [10] is subsumed in ours. In the context of concave games—that is, continuous games with individually concave payoff functions—Mertikopoulos & Zhou [34] showed that sharp global equilibria enjoy comparable convergence guarantees under FTRL with a vanishing step-size. While such step-size schedules are effective at suppressing noise in the long run, they do so at the cost of significantly slowing down the algorithm's convergence. By contrast, we focus on fast, *constant* step-size schedules, which are widely used in practice due to their simplicity and often superior empirical performance. In this regime, we show that entropically regularized learning with a constant step-size converges to robust equilibria at a geometric rate, compared to distinctly subgeometric rates in the case of vanishing step-size policies—subsuming in this way a range of previous results for finite [19] and stochastic games [20].

## 2 Preliminaries

We start by briefly reviewing some basics of game theory and regularized learning, introducing the necessary context for our results.

**2.1. The game-theoretic framework.** Throughout our paper, we focus on a class of *continuous games* consisting of a finite set of players $i \in \mathcal{N} = \{1, \dots, N\}$, and defined by the following primitives:

1. Each player $i \in \mathcal{N}$ has access to a compact convex subset $\mathcal{X}_i$ of some finite dimensional vector space $\mathcal{V}_i$, describing the set of *actions* available to said player. By $\mathcal{X} := \prod_i \mathcal{X}_i$ we denote the space of all ensembles $x = (x_1, \dots, x_N)$ of actions $x_i \in \mathcal{X}_i$ that are independently chosen by each player $i \in \mathcal{N}$. We will also write $x = (x_i; x_{-i})$ to emphasize the action of player $i \in \mathcal{N}$ against the joint action profile $x_{-i} \equiv (x_j)_{j \neq i}$ of all other players.

2. The players' rewards are determined by their individual *payoff functions* $u_i \colon \mathcal{X} \to \mathbb{R}$, assumed to be continuously differentiable for all $i \in \mathcal{N}$. Denoting by $\mathcal{Y}_i \equiv \mathcal{V}_i^*$ the dual space of $\mathcal{V}_i$, we define

the individual gradient vector $v_i \colon \mathcal{X} \to \mathcal{Y}_i$ of player $i \in \mathcal{N}$ by

$$v_i(x) = \nabla_{x_i} u_i(x_i; x_{-i}) \tag{1}$$

and the ensemble $v(x) = (v_1(x), \ldots, v_N(x)) \in \mathcal{Y} \equiv \prod_{i \in \mathcal{N}} \mathcal{Y}_i$ thereof.

A *continuous game* is then defined as a tuple $\mathcal{G} \equiv \mathcal{G}(\mathcal{N}, \mathcal{X}, u)$ with players, actions and payoff functions as above.

**Nash equilibrium.**    The best known solution concept in game theory is that of a *Nash equilibrium* (NE), which characterizes the actions $x^* \in \mathcal{X}$ from which no player has incentive to unilaterally deviate. Formally, $x^* \in \mathcal{X}$ is a Nash equilibrium if

$$u_i(x^*) \geq u_i(x_i; x_{-i}^*) \quad \text{for all } x_i \in \mathcal{X}_i, i \in \mathcal{N}. \tag{NE}$$

A game $\mathcal{G} \equiv \mathcal{G}(\mathcal{N}, \mathcal{X}, u)$ always admits a Nash equilibrium if $\mathcal{X}$ is compact and each player's payoff function $u_i$ is *individually concave* in the sense that $u_i(x_i; x_{-i})$ is concave in $x_i$ for all $x_{-i} \in \mathcal{X}_{-i}$ [14, 44]. In this case, basic arguments from convex analysis [42, 43] show that $x^*$ is an equilibrium of $\mathcal{G}$ if and only if it satisfies the (Stampacchia) variational inequality

$$\langle v(x^*), x - x^* \rangle \leq 0 \quad \text{for all } x \in \mathcal{X}. \tag{VI}$$

If the players' functions are not individually concave, a game may not admit a Nash equilibrium. In that case, it is more meaningful to consider *local* Nash equilibria, i.e., profiles $x^* \in \mathcal{X}$ such that

$$u_i(x^*) \geq u_i(x_i; x_{-i}^*) \quad \text{for all } x \text{ in a neighborhood } \mathcal{U} \text{ of } x^* \text{ in } \mathcal{X}. \tag{LNE}$$

In stark contrast to games with individually concave payoff functions, (VI) no longer characterizes local Nash equilibria: specifically, by first-order stationarity, we have (LNE) $\implies$ (VI) but the converse need not hold; in fact, a solution $x^*$ of (VI) may be a global payoff *maximizer* for all $i \in \mathcal{N}$.
*Note.* In the sequel, we will work with general continuous games that may not admit a global equilibrium—but admit *local* Nash equilibria. To streamline our presentation, we will use the term "equilibrium" without any further qualification to refer to *local* equilibria, and we will say explicitly "global equilibria" for profiles satisfying (NE).

### 2.2. Regularized learning in games.
The most widely used framework for learning in games, is the so called *"follow the regularized leader"* (FTRL) template, primarily because it leads to no regret in a wide variety of settings [48, 49]. The corresponding update rule hinges on the notion of a *regularized best response*, and proceeds as

$$y_{t+1} = y_t + \gamma \hat{v}_t, \qquad x_t = Q(y_t) \qquad \text{for } t = 1, 2, \ldots \tag{FTRL}$$

where $(i)$ $x_t \in \mathcal{X}$ denotes the players' action profile at step $t$; $(ii)$ $y_t = (y_{i,t})_{i \in \mathcal{N}} \in \mathcal{Y}$ is an auxiliary process that aggregates historical feedback into a compact state representation, i.e., a proxy for the players' empirical performance up to time $t$; $(iii)$ $\hat{v}_t = (\hat{v}_{i,t})_{i \in \mathcal{N}} \in \mathcal{Y}$ denotes the current gradient-like payoff signal; $(iv)$ $\gamma > 0$ is the learning rate, or step-size parameter of the process; and $(v)$ $Q \colon \mathcal{Y} \to \mathcal{X}$ is a mapping between the auxiliary process on the dual space $\mathcal{Y}$, and the players' strategy space $\mathcal{X}$. In what follows, we analyze the key components of this framework.

**The algorithm's step-size.**    Throughout this work, we adopt a *constant* step-size routine. This stands in contrast to the stochastic approximation literature [8, 11, 29], where (FTRL) is typically implemented with a *vanishing* step-size satisfying the Robbins-Monro summability conditions $\sum_t \gamma_t = \infty$, $\sum_t \gamma_t^2 < \infty$ [41], which is known to promote convergence by gradually suppressing the effect of noise [34].

On the other hand, in practical applications, it is common to employ a *constant* (or non-diminishing) step-size for several reasons. First, constant step-sizes are easier to tune and maintain, making them more suitable for large-scale or production environments. Moreover, methods with vanishing step-sizes often experience long warm-up phases and converge slowly to a neighborhood of the equilibrium. In comparison, constant step-size methods in machine learning settings typically reach the vicinity of a solution much faster—often within 0.1% accuracy [16]. Indeed, many state-of-the-art architectures, including transformers and large language models, use step-size schedules that remain effectively constant over billions or even trillions of samples [15].

**The mirror map.** A central ingredient of regularized learning is the mirror map $Q \equiv (Q_i)_{i \in \mathcal{N}}$, with each $Q_i \colon \mathcal{Y}_i \to \mathcal{X}_i$ induced by a strongly convex *regularizer* $h_i \colon \mathcal{X}_i \to \mathbb{R}$ that promotes stability during the learning process. To streamline our presentation and letting $h(x) = \sum_{i \in \mathcal{N}} h_i(x_i)$, the players' *mirror map* is defined as

$$Q(y) := \arg\max_{x \in \mathcal{X}} \{ \langle y, x \rangle - h(x) \} \tag{2}$$

In the rest of our paper, we will write $\mathcal{X}_h = \operatorname{im} Q$ for the image of $\mathcal{Y}$ under $Q$—and, likewise, $\mathcal{X}_{h_i} = \operatorname{im} Q_i$ for each player $i \in \mathcal{N}$. In particular, if $Q$ is interior-valued—that is, $\mathcal{X}_h = \operatorname{ri} \mathcal{X}$—we will say that $h$ is *steep* because, in this case, the (sub)gradients of $h$ explode to infinity as $x \to \operatorname{bd} \mathcal{X}$ (i.e., $h$ becomes "infinitely steep"); instead, if $\operatorname{im} Q = \mathcal{X}$, we will say that $h$ is *non-steep*. For a detailed discussion on this distinction and related concepts, see Appendix A.

Different choices of the regularizer $h$ induce different projection-like operations, adapted to the geometry of the underlying space. We describe two mainstay examples below.

**Example 2.1** (Euclidean projection). The quadratic regularizer $h(x) = \|x\|_2^2/2$ gives rise to the Euclidean projection $Q(y) = \operatorname{proj}_{\mathcal{X}}(y) = \arg\min_{x \in \mathcal{X}} \|y - x\|_2$. In this case, $h$ is non-steep. ☕

**Example 2.2** (Exponential weights). For $\mathcal{A}_i$ a *finite* set of actions per player $i \in \mathcal{N}$, and $\mathcal{X}_i \equiv \Delta(\mathcal{A}_i)$, the entropic regularizer $h_i(x_i) = \sum_{\alpha_i \in \mathcal{A}_i} x_{i\alpha_i} \log x_{i\alpha_i}$ gives rise to the logit map, defined via $Q_i(y_i) = \exp(y_i)/\|\exp(y_i)\|_1$, where $\exp(y_i)$ denotes the element-wise exponential of $y_i$. ☕

**The feedback process.** Throughout this work, we consider two distinct feedback models: (*i*) *stochastic gradients*; and (*ii*) *payoff-based* feedback. We describe both frameworks below.

**Stochastic gradient feedback.** At every time step $t$, each player $i \in \mathcal{N}$ has access to a *stochastic first-order oracle* (SFO)—that is, a noisy version of their individual gradient vector of the form:

$$\hat{v}_t = v(x_t) + U_t \quad \text{with} \quad \mathbb{E}[U_t \mid \mathcal{F}_t] = 0 \tag{SFO}$$

where $U_t$ is zero-mean and conditionally sub-Gaussian given the information $\mathcal{F}_t$ generated up to time $t \in \mathbb{N}$. In other words, players observe unbiased estimates of their individual gradient vectors.

**Payoff-based feedback.** Unlike the (SFO) model where players have access to a black-box oracle that provides noisy gradient information, it is often more realistic to consider a payoff-based paradigm where players observe *only* their realized payoffs—that is, a single scalar value—and have to reconstruct an estimate of their individual gradient vectors.

The most widely used method in this setting is the *single-point stochastic approximation* (SPSA) framework of [12, 50], which is based on finite differences along randomly sampled directions. Specifically, denoting the set of unit directions $\mathcal{E}_i := \{\pm e_1, \ldots, \pm e_{d_i}\}$ that span the affine hull of $\mathcal{X}_i$ of dimension $d_i$, each player $i \in \mathcal{N}$ draws a direction $w_{i,t} \in \mathcal{E}_i$ uniformly at random in every round $t \in \mathbb{N}$. Since the perturbed action $x_{i,t} + \varepsilon_t w_{i,t}$ may lie outside $\mathcal{X}_i$ for a perturbation radius $\varepsilon_t > 0$, we introduce a pivot element $p_i \in \operatorname{ri}(\mathcal{X}_i)$ and a radius $r_i > \varepsilon_t$ such that $p_i + r_i w_i \in \mathcal{X}_i$ for all $w_i \in \mathcal{E}_i$. Based on these, we define the feasibility-adjusted action $x_{i,t}^{\varepsilon} := x_{i,t} + (\varepsilon_t/r_i)(p_i - x_{i,t}) \in \mathcal{X}_i$. Finally, each player queries the perturbed action $\hat{x}_{i,t} \equiv x_{i,t}^{\varepsilon} + \varepsilon_t w_{i,t}$ which is an element of $\mathcal{X}_i$, and observes the realized payoff value $u_i(\hat{x}_t)$.[2] The gradient vector is, then, estimated via the single-point stochastic approximation scheme:

$$\hat{v}_{i,t} := (d_i/\varepsilon_t) \, u_i(\hat{x}_t) \, w_{i,t} \tag{SPSA}$$

Importantly, the feasibility adjustment ensures that the perturbed action $\hat{x}_t$ remains within the players' action set $\mathcal{X}$, while preserving the direction of the original perturbation $w_t$. As we show in Appendix A, (SPSA) enjoys the bounds

$$\|\mathbb{E}[\hat{v}_t \mid \mathcal{F}_t] - v(x_t)\|_* = \mathcal{O}(\varepsilon_t) \quad \text{and} \quad \|\hat{v}_t\|_* = \mathcal{O}(1/\varepsilon_t) . \tag{3}$$

These statistical properties of (SPSA) will play a crucial role in establishing its convergence guarantees; we will revisit them in Section 4.

---

[2]Since $r_i > \varepsilon_t$, we write $x_{i,t}^{\varepsilon} = x_{i,t}(1 - \varepsilon_t/r_i) + (\varepsilon_t/r_i)p_i$ which is a convex combination of points in $\mathcal{X}_i$. Regarding $\hat{x}_{i,t}$, note it can be written as $\hat{x}_{i,t} = x_{i,t}(1 - \varepsilon_t/r_i) + (\varepsilon_t/r_i)(p_i + r_i w_{i,t})$, which is also a convex combination of points in $\mathcal{X}_i$. Thus, both belong to $\mathcal{X}_i$.

# 3 Strategic robustness: Geometric and variational characterization

We begin in this section by addressing the strategic aspects of the equilibrium robustness question, namely:

*Which equilibria remain invariant under small—but otherwise arbitrary—perturbations of the game?*

We take this desideratum as the starting point for our definition of *strategic robustness*, that is, action profiles that remain (local) equilibria under small disturbances in the underlying game. This leads to a delicate interplay between the variational and geometric aspects of the underlying game, which we detail below.

**3.1. A first approach and insights.** The first step in our analysis is to quantify the meaning of "small". In this regard, a natural way to measure the distance between two concave games, $\mathcal{G} \equiv \mathcal{G}(\mathcal{N}, \mathcal{X}, u)$ and $\tilde{\mathcal{G}} \equiv \mathcal{G}(\mathcal{N}, \mathcal{X}, \tilde{u})$, would be via the uniform distance

$$\rho(\mathcal{G}, \tilde{\mathcal{G}}) := \max_{i \in \mathcal{N}} \sup_{x \in \mathcal{X}} |u_i(x) - \tilde{u}_i(x)|. \tag{4}$$

Intuitively, if this quantity is small enough, the two games are nearly indistinguishable from a strategic perspective, since for every strategy profile $x \in \mathcal{X}$, the payoffs in $\mathcal{G}$ and $\tilde{\mathcal{G}}$ are almost the same. Thus, one might expect that at least *some* equilibria of $\mathcal{G}$ should persist under sufficiently small perturbations, especially given that a Nash equilibrium is defined in terms of the game's payoff functions themselves.

Perhaps surprisingly, as we show below, this definition of distance *cannot* provide a meaningful concept of equilibrium robustness.

**Proposition 1.** *For any game $\mathcal{G}$ and any equilibrium $x^* \in \mathcal{X}$ of $\mathcal{G}$, there exists a perturbed game $\tilde{\mathcal{G}}$, arbitrarily close to $\mathcal{G}$ in the uniform metric* (4) *such that $x^* \in \mathcal{X}$ is not an equilibrium of $\tilde{\mathcal{G}}$.*

To show this, we provide Examples 3.1 and 3.2 which, taken together, cover all possible types of equilibria in continuous games in the sense of (VI).

**Example 3.1.** Let $\mathcal{G}$ be a continuous game, and let $x^* \in \mathcal{X}$ be an equilibrium of $\mathcal{G}$ such that $\langle v_i(x^*), p_i - x_i^* \rangle < 0$ for some player $i \in \mathcal{N}$ and $p_i \in \mathcal{X}_i$. For arbitrary $\varepsilon > 0$, define $\tilde{u}_i : \mathcal{X} \to \mathbb{R}$ as

$$\tilde{u}_i(x) := u_i(x) - \varepsilon \exp\left(2\varepsilon^{-1}\langle v_i(x^*), x_i - x_i^* \rangle\right) \tag{5}$$

which is a continuously differentiable concave function in $x_i$, and let $\tilde{u}_j \equiv u_j$ for all $j \neq i$, $j \in \mathcal{N}$. Since $x^* \in \mathcal{X}$ is an equilibrium of $\mathcal{G}$, it holds $\langle v_i(x^*), x_i - x_i^* \rangle \leq 0$ for all $x_i \in \mathcal{X}_i$, which implies that

$$\rho(\mathcal{G}, \tilde{\mathcal{G}}) = \sup_{x \in \mathcal{X}} |u_i(x) - \tilde{u}_i(x)| = \varepsilon \sup_{x_i \in \mathcal{X}_i} \exp\left(2\varepsilon^{-1}\langle v_i(x^*), x_i - x_i^* \rangle\right) = \varepsilon. \tag{6}$$

Computing the individual gradient vector of player $i \in \mathcal{N}$, we obtain

$$\tilde{v}_i(x) = v_i(x) - 2v_i(x^*) \exp\left(2\varepsilon^{-1}\langle v_i(x^*), x_i - x_i^* \rangle\right) \tag{7}$$

and, evaluating it at $x^* \in \mathcal{X}$, we get, $\tilde{v}_i(x^*) = -v_i(x^*)$. Therefore, for $x = (p_i; x_{-i}^*) \in \mathcal{X}$, we have

$$\langle \tilde{v}(x^*), x - x^* \rangle = -\langle v_i(x^*), p_i - x_i^* \rangle > 0 \tag{8}$$

i.e., $x^* \in \mathcal{X}$ is not an equilibrium point of the perturbed game $\tilde{\mathcal{G}}$. ❧

**Example 3.2.** Let $\mathcal{G}$ be a continuous game, and let $x^* \in \mathcal{X}$ be an equilibrium of $\mathcal{G}$ such that $\langle v(x^*), x - x^* \rangle = 0$ for all $x \in \mathcal{X}$. Fix a player $i \in \mathcal{N}$ and $p_i \in \mathcal{X}_i$, and let $y_i \in \mathcal{V}_i^*$ with $\langle y_i, p_i - x_i^* \rangle > 0$. For arbitrary $\varepsilon > 0$, let $\tilde{u}_i : \mathcal{X} \to \mathbb{R}$ be defined as

$$\tilde{u}_i(x) := u_i(x) + \varepsilon \operatorname{diam}(\mathcal{X}_i)^{-1} \|y_i\|_*^{-1} \langle y_i, x_i - x_i^* \rangle \tag{9}$$

which is a concave function in $x_i$, and $\tilde{u}_j \equiv u_j$ for all $j \neq i$, $j \in \mathcal{N}$. Then, we readily get that

$$\rho(\mathcal{G}, \tilde{\mathcal{G}}) = \sup_{x \in \mathcal{X}} |u_i(x) - \tilde{u}_i(x)| = \varepsilon \operatorname{diam}(\mathcal{X}_i)^{-1} \|y_i\|_*^{-1} \sup_{x_i \in \mathcal{X}_i} |\langle y_i, x_i - x_i^* \rangle| \leq \varepsilon. \tag{10}$$

Computing the individual gradient vector of player $i \in \mathcal{N}$, we obtain

$$\tilde{v}_i(x) = v_i(x) + \varepsilon \operatorname{diam}(\mathcal{X}_i)^{-1} \|y_i\|_*^{-1} y_i \tag{11}$$

Therefore, for $x = (p_i; x_{-i}^*) \in \mathcal{X}$, it holds by the example's assumptions that

$$\langle \tilde{v}(x^*), x - x^* \rangle = \langle v_i(x^*), p_i - x_i^* \rangle + \varepsilon \operatorname{diam}(\mathcal{X}_i)^{-1} \|y_i\|_*^{-1} \langle y_i, p_i - x_i^* \rangle > 0 \tag{12}$$

where we used that $\langle v_i(x^*), p_i - x_i^* \rangle = 0$ and $\langle y_i, p_i - x_i^* \rangle > 0$, as per our original assumptions. Thus, $x^* \in \mathcal{X}$ is not an equilibrium of the perturbed game $\tilde{\mathcal{G}}$. ❧

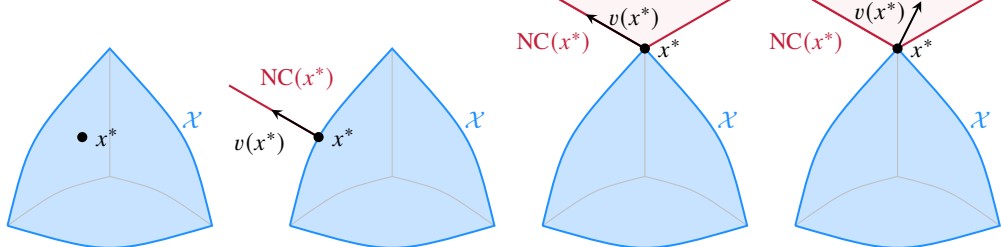

**Figure 1:** Different equilibrium configurations: an interior equilibrium ($v(x^*) = 0$); a boundary, non-extreme equilibrium (normal cone with empty topological interior); an extreme, non-robust equilibrium ($v(x^*)$ on the boundary of the normal cone); a robust equilibrium ($v(x^*)$ in the interior of the normal cone). Only the robust equilibrium remains invariant under strategic perturbations of the underlying game.

*Remark* 1. In Examples 3.1 and 3.2, if $\mathcal{G}$ is concave, so is $\tilde{\mathcal{G}}$, indicating that this notion of distance is not proper even within the class of concave games.

The preceding examples demonstrate that under the distance (4), even an arbitrarily small perturbation to the payoff function of a *single* player can destroy *any* equilibrium.[3] This phenomenon arises because, although an equilibrium is defined in terms of payoff functions, the first-order stationarity condition in (VI) shows that it fundamentally depends on the individual gradient vectors. Therefore, any meaningful notion of distance between two games must likewise be aware of the behavior of the individual gradient vectors.

**3.2. Defining strategic robustness.** As illustrated in Examples 3.1 and 3.2, small changes in the payoffs, though negligible in the uniform norm, can alter the equilibrium landscape quite significantly. To address this, we refine the notion of distance between games $\mathcal{G}$ and $\tilde{\mathcal{G}}$ as follows:

$$\mathrm{dist}\big(\mathcal{G}, \tilde{\mathcal{G}}\big) \coloneqq \sup_{x \in \mathcal{X}} \|v(x) - \tilde{v}(x)\|_* \tag{13}$$

With this definition in hand, we are now ready to state the concept of strategic robustness in the class of continuous games.

**Definition 1.** An equilibrium $x^* \in \mathcal{X}$ of a game $\mathcal{G}$ is called *strategically robust* if there exists $\varepsilon > 0$ such that for *any* game $\tilde{\mathcal{G}}$ with $\mathrm{dist}\big(\mathcal{G}, \tilde{\mathcal{G}}\big) < \varepsilon$, $x^*$ is also an equilibrium of $\tilde{\mathcal{G}}$.

As we explore next, this definition offers a meaningful notion of "closeness" for equilibrium stability, one that is grounded not in the payoff values themselves, but in the geometry they induce.

**Geometric characterization.** To provide a geometric characterization, we zoom in on the variational structure that governs Nash equilibria. Specifically, we show that strategically robust equilibria $x^* \in \mathcal{X}$ are precisely those solutions of (VI) for which the inequality is strict for all feasible deviations. Formally, we have the following characterization:

**Theorem 1.** *Let $x^* \in \mathcal{X}$ be a joint action profile in $\mathcal{G}(\mathcal{N}, \mathcal{X}, u)$. Then the following are equivalent:*

(i) *$x^*$ is a strategically robust equilibrium.*

(ii) *$\langle v(x^*), z \rangle \leq -m\|z\|$ for some $m > 0$ and all $z \in \mathrm{TC}(x^*)$, where $\mathrm{TC}(x^*)$ is the closure of all rays emanating from $x^*$ and intersecting $\mathcal{X}$ in at least one other point.*

(iii) *$v(x^*) \in \mathrm{int}(\mathrm{PC}(x^*))$, where $\mathrm{PC}(x^*) \coloneqq \{y \in \mathcal{Y} : \langle y, z \rangle \leq 0, \text{ for all } z \in \mathrm{TC}(x^*)\}$.*

Intuitively, Theorem 1 suggests that strategically robust equilibria are precisely those points $x^* \in \mathcal{X}$ where the associated gradient vector $v(x^*)$ lies in the topological interior of the polar cone $\mathrm{PC}(x^*)$, i.e.,

$$\langle v(x^*), z \rangle < 0 \quad \text{for all } z \in \mathrm{TC}(x^*). \tag{14}$$

We thus conclude that strategic robustness can only occur at boundary points where the tangent cone is pointed; if the feasible set is locally flat at $x^* \in \mathcal{X}$, the corresponding polar cone has empty

---

[3]Such variations are not possible in the class of finite games, so, in this much more restrictive class, the sup-norm of the payoff differences is a valid metric to measure robustness.

interior, and robustness is not possible. This phenomenon is illustrated in Fig. 1, and the full proof of Theorem 1 is provided in Appendix B.

*Remark* 2. Both Examples 3.1 and 3.2 violate the condition in Definition 1, but for different reasons. In the former case, although the perturbed payoffs can be made arbitrarily close to the original, the perturbed gradient vector at equilibrium can become arbitrarily large, making the distance $\text{dist}(\mathcal{G}, \tilde{\mathcal{G}})$ exceed any $\varepsilon > 0$. In the latter case, the polar cone $\text{PC}(x^*)$ at the equilibrium has empty interior, so strategic robustness cannot hold at $x^*$. ☙

*Remark* 3. Several important classes of games admit robust equilibria for all but a measure-zero set of instances. Examples include nonatomic, non-splittable routing games with arbitrary increasing cost functions [45], Markov potential games arising in multi-agent reinforcement learning [32], Cournot oligopolies in which firms have no or limited price-setting power [37], etc.

In the next section, we examine the dynamic implications of this result by studying the robustness of such equilibria under (FTRL).

# 4 From strategic to dynamic robustness: Convergence results

So far, we focused on strategic robustness, a static notion determined solely by the underlying game and the local geometry around the equilibrium in question. In this section, we shift to the dynamic perspective of our central question and explore which equilibria admit robust convergence guarantees, namely, equilibria that can emerge as stable outcomes of regularized learning under feedback and initialization uncertainty, regardless of the specific choice of regularizer.

To this end, we first establish that non-equilibrium points cannot arise as limits of (FTRL), even with *perfect* gradient feedback. Formally, we have the following proposition, whose proof is provided in Appendix C.

**Proposition 2.** *Suppose that* (FTRL) *is run with perfect gradient feedback of the form* $\hat{v}_t = v(x_t)$ *for all* $t = 1, 2, \ldots$, *and assume that* $x_t$ *converges to some* $\hat{x} \in \mathcal{X}$. *Then* $\hat{x}$ *is an equilibrium of* $\mathcal{G}$.

Having excluded non-equilibrium points as positive probability outcomes of a learning process, we now turn to identifying equilibria that are robust from a dynamic standpoint, and more precisely, under that of (FTRL). In this regard, strategically robust equilibria serve as natural candidates, as their stability with respect to game perturbations suggests they may also admit robust convergence guarantees. This is further supported by the finding that equilibrium points in the interior of the strategy space $\mathcal{X}$ cannot be limit points: in particular, we show below that, even under i.i.d. stochastic noise, the iterates of (FTRL) diverge from such equilibria almost surely.

**Proposition 3.** *Let* $x^* \in \text{ri}(\mathcal{X})$ *be a Nash equilibrium of* $\mathcal{G}(\mathcal{N}, \mathcal{X}, u)$, *and* $(x_t)_{t \in \mathbb{N}}$ *be the sequence of play induced by* (FTRL) *with* $\hat{v}_t = v(x_t) + U_t$, *where* $U_t$ *i.i.d. with* $\mathbb{E}[U_t] = 0$ *and* $\text{cov}(U_t) \succ 0$ *for all* $t \in \mathbb{N}$. *Then:*

$$\mathbb{P}\left(\lim_{t \to \infty} x_t = x^*\right) = 0 \qquad \text{for any } x_1 \in \mathcal{X}_h. \tag{15}$$

*Remark* 4. The condition $\text{cov}(U_t) \succ 0$ is not necessary. In fact, it suffices to have $\text{cov}(U_t)$ non-degenerate in a direction $p - x^*$ for $p \in \mathcal{X}$, but we state our result under stronger assumptions for simplicity. ☙

The key idea of the proof, which is deferred to Appendix C, is that, since $x^* \in \text{ri}(\mathcal{X})$, we have $\langle v(x^*), x - x^* \rangle = 0$ for all $x \in \mathcal{X}$. At the same time, as $\text{cov}(U_t) \succ 0$, the quantity $\langle U_t, x - x^* \rangle$ fluctuates and remains bounded away from zero infinitely often, thereby preventing convergence.

**4.1. Learning with gradient-based feedback.** In view of the impossibility result of Proposition 3, we shift our focus on the convergence of (FTRL) toward strategically robust equilibria. We first consider the gradient feedback model, where each player receives an unbiased estimate of their individual gradient vector via (SFO). Specifically, we analyze the behavior of (FTRL) and we establish local convergence guarantees toward strategically robust equilibria with high probability. This is encoded in the following theorem:

**Theorem 2.** *Let* $x^* \in \mathcal{X}$ *be a strategically robust equilibrium of* $\mathcal{G}(\mathcal{N}, \mathcal{X}, u)$. *Fix a confidence level* $\delta > 0$, *and let* $(x_t)_{t \in \mathbb{N}}$ *be the iterates of* (FTRL) *with feedback provided by* (SFO)*, and step-size* $\gamma > 0$ *sufficiently small. Then, there exists a neighborhood* $\mathcal{U}$ *of* $x^*$ *in* $\mathcal{X}_h$ *such that:*

$$\mathbb{P}\left(\lim_{t \to \infty} x_t = x^*\right) \geq 1 - \delta \quad \text{if } x_1 \in \mathcal{U}. \tag{16}$$

Before proceeding, a few remarks are in order. Since continuous games may admit multiple Nash equilibria, global convergence guarantees are in general unattainable. As such, our analysis focuses on the local convergence landscape of (FTRL). As a sidenote, it is important to emphasize that the convergence result is robust to the choice of regularizer, relying solely on the general conditions outlined in Section 2 rather than any particular functional form. We outline below the main steps of the proof, with full details provided in Appendix C.

*Proof Sketch.* The key idea is that the auxiliary process $y_t$, which aggregates the players' gradient updates, diverges to infinity in a direction that steers the induced sequence $x_t = Q(y_t)$ toward the equilibrium in question. More formally, the proof relies on the following intuition. From Theorem 1 and the continuity of the players' payoffs, strategic robustness implies that, in a neighborhood of a robust equilibrium $x^*$, the players' individual gradient fields point toward $x^*$. Consequently, the process $y_t$ accumulates gradient steps that, on average, are aligned with the interior of the normal cone $\mathrm{NC}(x^*)$ to the action space $\mathcal{X}$ at $x^*$. As a result, $y_t$ exhibits a consistent drift that carries it deeper into a "copy" of the normal cone $\mathrm{NC}(x^*)$ embedded in the gradient space $\mathcal{Y}$. Moreover, we show that, with high probability, once $y_t$ enters this region, it remains there, provided that the algorithm is not initialized too far from $x^*$. Combining the above, Proposition C.1 establishes that the sequence of actions $x_t = Q(y_t)$ generated by (FTRL) converges to $x^*$. �

While our main focus lies on the qualitative convergence behavior of (FTRL), stronger guarantees can be obtained under additional structural assumptions on the strategy space and the regularizer. In particular, suppose that $\mathcal{X}$ is a polyhedral domain of the form $\mathcal{X} := \{x \in \mathbb{R}_+^d \mid Ax = b\}$ for some $A \in \mathbb{R}^{m \times d}$ and $b \in \mathbb{R}^m$, and $h$ is decomposable with kernel function $\theta$, i.e., $h$ can be written as $h(x) = \sum_{j=1}^d \theta(x_j)$ for some continuous function $\theta : \mathbb{R}_+ \to \mathbb{R}$ with locally Lipschitz $\theta''$ and $\theta'' > 0$. Under these conditions, we obtain the explicit convergence rates for the (FTRL) dynamics, as follows.

**Theorem 3.** *If, in addition, $\mathcal{X}$ is a polyhedral domain and $h$ is decomposable with kernel $\theta$, on the event $E := \{\lim_{t \to \infty} x_t = x^*\}$ it holds:*

$$\|x_t - x^*\| = \phi(-\Theta(t)) \tag{17}$$

*where $\phi$ is the rate function defined via*

$$\phi(z) := \begin{cases} (\theta')^{-1}(z) & \text{if } z > \theta'(0^+) \\ 0 & \text{if } z \le \theta'(0^+) \end{cases} \tag{18}$$

*Remark* 5. For the setting of Example 2.2, with $\mathcal{X} = \Delta(\mathcal{A})$, $\theta(z) = z \log z$ and $x^*$ a *strict* Nash equilibrium, the convergence rate of (FTRL) as per Theorem 3, becomes $\|x_t - x^*\| = \exp(-\Theta(t))$.

*Remark* 6. For *finite games*, [19] showed that under a step-size schedule of the form $\gamma_t \propto 1/t^p$, the Robins-Monro summability conditions require $p \in (1/2, 1]$, leading to convergence rates from $\phi(-\Theta(t^{1-p}))$ to $\phi(-\Theta(\log t))$. Our convergence guarantees remain valid under these step-size schedules, though they yield the slower aforementioned rates.

*Remark* 7. It is important to highlight the different behavior of (FTRL), often referred to as a "lazy" variant of mirror descent [22], with that of the mirror descent algorithm, defined via the update

$$x_{t+1} = Q(\nabla h(x_t) + \gamma \hat{v}_t) \qquad \text{for } t = 1, 2, \dots \tag{MD}$$

where $\nabla h(x)$ denotes a continuous selection of $\partial h(x)$ [12]. To illustrate the difference, consider the single-agent problem of maximizing $u(x) = x$ over $\mathcal{X} = [0, 1]$, where $x^* = 1$ is a robust equilibrium. Using the Euclidean regularizer (see Example 2.1), (MD) reduces to the projected gradient algorithm

$$x_{t+1} = \Pi(x_t + \gamma \hat{v}_t) \tag{SGA}$$

where $\hat{v}_t$ is a stochastic gradient of $u$ at $x_t$, i.e., $\hat{v}_t = 1 + U_t$ where $U_n$ is a Bernoulli process with $U_t = \pm 1$ with probability $1/2$. However, even if $x_t = x^*$ for some $t$, we then have that $x_{t+1} = 1 - \gamma$ with probability $1/2$. Thus, by a straightforward application of the Borel-Cantelli lemma, we conclude that, with probability 1, (SGA) does not converge to $x^*$. Through this toy example, note that although (MD) and (FTRL) use the same mirror map $Q$ to select actions, they differ fundamentally in how feedback is processed. The "eager" nature of (MD) makes it more sensitive to noise, whereas (FTRL) maintains a cumulative dual variable $y_t$ that aggregates all past feedback, effectively smoothing out fluctuations over time. Also, note that for steep regularizers, the iterations (MD) and (FTRL) coincide, as the mirror map $Q$ is essentially injective (see Appendix A for more details). Therefore, differences in their behavior arise in the case where steepness does not hold.

**4.2. Learning with payoff-based feedback.** We now turn to the payoff-based feedback model, where players observe only their realized payoffs and use them to estimate gradients indirectly via (SPSA). This feedback model introduces higher variance and structural bias due to the diminishing sampling radius and the feasibility corrections. Nevertheless, we show that strategically robust equilibria retain their dynamic robustness: they still locally attract the (FTRL) dynamics with high probability, as the following theorem suggests.

**Theorem 4.** *Let $x^* \in \mathcal{X}$ be a strategically robust equilibrium of $\mathcal{G}$. Fix a confidence level $\delta > 0$, and let $(x_t)_{t \in \mathbb{N}}$ be the iterates of (FTRL) run with (SPSA) with $\varepsilon_t \propto 1/t^p$ for some $p \in (0, 1/2)$ and step-size $\gamma > 0$ sufficiently small. Then, there exists a neighborhood $\mathcal{U}$ of $x^*$ such that:*

$$\mathbb{P}\left(\lim_{t \to \infty} x_t = x^*\right) \geq 1 - \delta \qquad \text{if } x_1 \in \mathcal{U}. \tag{19}$$

*If, in addition, $\mathcal{X}$ is affinely constrained and $h$ is decomposable with kernel $\theta$, then, whenever $x_t$ converges to $x^*$, we have:*

$$\|x_t - x^*\| = \phi(-\Theta(t)). \tag{20}$$

Despite the scarcity of information inherent in the payoff-based feedback model, strategically robust equilibria retain not only their convergence properties but also their convergence speed, under the additional structural assumptions on the regularizer and domain, matching that of the (SFO) feedback setting. This is further discussed along with the proof of the theorem in Appendix C.

**4.3. Convergence landscape beyond strategic robustness.** Having established the robust convergence properties of strategically robust equilibria, a natural question arises: *Can we expect robust convergence guarantees toward equilibria that lack this structural property?* As we show below, the answer is not encouraging: strategic robustness is essentially necessary for robust convergence.

To make this limitation precise, we move beyond the interior of the strategy space, where Proposition 3 rules out equilibria as potential limit points, and shift our focus to non-robust equilibria on the boundary. To illustrate the behavior of (FTRL) in this setting, we construct a game with a unique equilibrium that exhibits fundamentally different long-run behavior depending on the regularizer.

**Proposition 4.** *Consider the 1-player game $\mathcal{G}$ with $\mathcal{X} = [0, 1]$, $u(x) = -\frac{3}{4}x^{4/3}$ and $x^* = 0$. Let $(x_t)_{t \in \mathbb{N}}$ be the iterates of (FTRL) with $\gamma < 1$, and $\hat{v}_t = v(x_t) + U_t$, where $U_t$ are i.i.d. standard normal random variables for all $t \in \mathbb{N}$. Then, for any initial condition $y_1 \in \mathbb{R}$, we have:*

   *(i) For $h(x) = x \log x$, it holds $\mathbb{P}(\lim_{t \to \infty} x_t = x^*) = 0$.*

   *(ii) For $h(x) = -2\sqrt{x}$, it holds $\mathbb{P}(\lim_{t \to \infty} x_t = x^*) = 1$.*

The core idea of the proof of Proposition 4 (which we present in detail in Appendix C) is to construct a process $z_t$ that dominates $y_t$. Importantly, the process $z_t$ can be then viewed as a random walk with a diminishing drift whose rate of decay depends on the choice of regularizer. Depending on the magnitude of this drift, the process exhibits two sharply contrasting long-term behaviors: if the drift decays sufficiently fast, the process behaves like a zero-mean random walk and returns infinitely often with probability 1 (recurrence); conversely, if the drift diminishes at a slower rate, the process behaves like a random walk with constant drift and escapes to infinity with probability 1 (transience).

In view of the above, we conclude that strategic robustness cannot be relaxed without compromising convergence guarantees, even when the equilibrium lies on the boundary of the game's action space.

## 5   Concluding remarks

Our aim in this paper was to examine the robustness of Nash equilibria in continuous games, under both strategic and dynamic uncertainty. From a strategic standpoint, the notion of *strategic robustness* characterizes those (local) equilibria which remain invariant under small perturbations of the underlying game, and we derived a tight geometric characterization thereof in terms of the variational geometry of the game. From a dynamic standpoint, we focused on the stability of regularized learning under uncertainty, and we established a deep structural connection between the two notions. Strategic robustness guarantees dynamic robustness under (FTRL), and this implication is essentially tight: without strategic robustness, dynamic robustness cannot be ensured. To the best of our knowledge, this is the first study of its kind for continuous games, and we believe that this two-way implication elucidates the delicate interplay between static and dynamic considerations.

## Acknowledgments and Disclosure of Funding

Jose Blanchet gratefully acknowledges support from the Department of Defense through the Air Force Office of Scientific Research (Award FA9550-20-1-0397) and the Office of Naval Research (Grant 1398311), as well as from the National Science Foundation (Grants 2229012, 2312204, and 2403007). Panayotis Mertikopoulos is also a member of Archimedes/Athena RC and acknowledges financial support by the French National Research Agency (ANR) in the framework of the PEPR IA FOUNDRY project (ANR-23-PEIA-0003), the project IRGA-SPICE (G7H-IRG24E90), and project MIS 5154714 of the National Recovery and Resilience Plan Greece 2.0, funded by the European Union under the NextGenerationEU Program.

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

# A    Auxiliary results

As a preamble to our analysis, we provide some basic properties of the regularizers and the mirror maps, and present some auxiliary results from martingale theory and Markov processes that we will use throughout the sequel.

**A.1.  Mirror maps and results from convex analysis.** In this section, we provide a more detailed discussion of key notions from convex analysis, including mirror maps and regularizers.

To begin, let $(\mathcal{V}, \|\cdot\|)$ be a finite-dimensional normed vector space. Its dual space is denoted by $(\mathcal{V}^*, \|\cdot\|_*)$, where the dual norm is defined as

$$\|y\|_* \equiv \max\{\langle y, x \rangle : \|x\| \leq 1\}, \tag{A.1}$$

and $\langle y, x \rangle$ denotes the canonical pairing between $y \in \mathcal{V}^*$ and $x \in \mathcal{V}$. To maintain consistency with the notation used throughout the paper, we will refer to $\mathcal{V}^*$ as $\mathcal{Y}$ from this point onward.

Given a closed convex set $\mathcal{X} \subseteq \mathcal{V}$ and a point $p \in \mathcal{X}$, we define the tangent cone $\mathrm{TC}(p)$ and the polar cone $\mathrm{PC}(p)$ as follows:

$$\mathrm{TC}(p) = \mathrm{cl}\{z \in \mathcal{V} : p + tz \in \mathcal{X} \text{ for some } t > 0\} \tag{A.2}$$

and

$$\mathrm{PC}(p) = \{y \in \mathcal{Y} : \langle y, z \rangle \leq 0, \text{ for all } z \in \mathrm{TC}(p)\} \tag{A.3}$$

For a strongly convex regularizer $h : \mathcal{X} \to \mathbb{R}$, the *subdifferential* of $h$ at $x \in \mathcal{X}$ is defined as

$$\partial h(x) := \{y \in \mathcal{Y} : h(x') \geq h(x) + \langle y, x' - x \rangle \text{ for all } x' \in \mathcal{X}\} \tag{A.4}$$

and we denote the *domain of subdifferentiability* of $h$ as

$$\mathcal{X}_h = \{x \in \mathcal{X} : \partial h(x) \neq \varnothing\}. \tag{A.5}$$

In addition, the mirror map $Q$, defined via

$$Q(y) = \arg\max_{x \in \mathcal{X}} \{\langle y, x \rangle - h(x)\} \tag{A.6}$$

is single-valued on $\mathcal{Y}$, since the maximization problem admits a unique solution, as $h$ is strongly convex. Finally, by the optimality conditions of (A.6), we get that

$$x = Q(y) \quad \text{if and only if} \quad y \in \partial h(x). \tag{A.7}$$

since $0 \in y - \partial h(x)$. This readily implies that $\mathcal{X}_h = \operatorname{im} Q$. In general, we have

$$\mathrm{ri}(\mathcal{X}) \subseteq \mathcal{X}_h \subseteq \mathcal{X}, \tag{A.8}$$

where the first inclusion follows from standard results on the subdifferentiability of convex functions [42, Chap. 26], whereas the second is immediate from the definition of $\mathcal{X}_h$. This leads to two contrasting regimes: (*i*) $\mathcal{X}_h = \mathrm{ri}(\mathcal{X})$, in which case $h$ is called *steep*; and (*ii*) $\mathcal{X}_h = \mathcal{X}$, in which case $h$ is called *non-steep*.

Finally, we include here for future reference an elementary result concerning solid (convex) cones.

**Lemma A.1.** *Let $\mathcal{K}$ be a convex cone in $\mathbb{R}^d$ with nonempty topological interior, and let $z \in \mathrm{int}(\mathcal{K})$. Then there exists a finitely generated cone $\mathcal{K}'$ such that $z \in \mathrm{int}\,\mathcal{K}' \subseteq \mathrm{int}\,\mathcal{K}$.*

*Remark.* We stress here that, by $\mathrm{int}\,\mathcal{K}$ we mean the topological interior of $\mathcal{K}$ (which is nonempty by assumption), not the relative interior $\mathrm{ri}\,\mathcal{K}$ thereof (whis is always nonempty).

*Proof.* Since $\mathcal{K}$ is closed and $z \in \mathrm{int}\,\mathcal{K}$, there exists a closed ball $\mathcal{B}$ centered at $z$, which is entirely contained in $\mathrm{int}\,\mathcal{K}$ (an immediate consequence of the fact that $z$ is well-separated from the boundary $\mathrm{bd}\,\mathcal{K}$ of $\mathcal{K}$). Since $\mathcal{B}$ is not contained in any lower-dimensional subspace of $\mathbb{R}^d$, it is possible to find inductively $d$ linearly independent vectors $z_1, \ldots, z_d \in \mathcal{B}$ on the boundary $\mathrm{bd}\,\mathcal{B}$ of $\mathcal{B}$ such that $z$ is contained in the convex hull $\Delta(z_1, \ldots, z_d)$ (and, in particular, in the relative interior thereof). Thus, letting $\mathcal{K}' \equiv \mathcal{K}(z_1, \ldots, z_d)$ be the polyhedral cone generated by $z_1, \ldots, z_d$, we have $\mathcal{K}' \subseteq \mathcal{K}$ and $z \in \mathrm{int}\,\mathcal{K}'$ by construction, and our proof is complete. ∎

**A.2. Statistical bounds and results from probability theory.** In this section, we provide some basic statistical bounds for (SPSA), and we present some results that we will use freely in the sequel. We start with our bounds for (SPSA), specifically:

**Proposition A.1.** *The estimator* (SPSA) *enjoys the following bounds:*

$$\|\mathbb{E}[\hat{v}_t \mid \mathcal{F}_t] - v(x_t)\|_* = \mathcal{O}(\varepsilon_t) \quad \text{and} \quad \|\hat{v}_t\|_* = \mathcal{O}(1/\varepsilon_t). \tag{A.9}$$

*Proof.* Letting $\zeta_t := \varepsilon_t(w_t + (p - x_t)/r)$, we write $\hat{x}_t = x_t + \zeta_t$, and we have for player $i \in \mathcal{N}$:

$$u_i(\hat{x}_t)w_{i,t} = u_i(x_t)w_{i,t} + \langle \nabla u_i(x_t), \zeta_t \rangle w_{i,t} + \int_0^1 \langle \nabla u_i(x_t + \tau\zeta_t) - \nabla u_i(x_t), \zeta_t \rangle d\tau w_{i,t} \tag{A.10}$$

Now, the middle term can be unfolded as

$$\langle \nabla u_i(x_t), \zeta_t \rangle w_{i,t} = \langle \nabla_i u_i(x_t), \zeta_{i,t} \rangle w_{i,t} + \sum_{j \neq i} \langle \nabla_j u_i(x_t), \zeta_{j,t} \rangle w_{i,t} \tag{A.11}$$

and, noting that $\mathbb{E}[w_{i,t} \mid \mathcal{F}_t] = 0$, we take conditional expectation, and we get:

$$\mathbb{E}[\langle \nabla u_i(x_t), \zeta_t \rangle w_{i,t} \mid \mathcal{F}_t] = \varepsilon_t \, \mathbb{E}[\langle \nabla_i u_i(x_t), w_{it} \rangle w_{i,t} \mid \mathcal{F}_t] = (\varepsilon_t / d_i) v_i(x_t) \tag{A.12}$$

and

$$\mathbb{E}[u_i(x_t) w_{i,t} \mid \mathcal{F}_t] = 0 \tag{A.13}$$

Therefore, we have:

$$\|\mathbb{E}[\hat{v}_{i,t} \mid \mathcal{F}_t] - v_i(x_t)\| = \left\| \mathbb{E}\left[ \int_0^1 \langle \nabla u_i(x_t + \tau \zeta_t) - \nabla u_i(x_t), \zeta_t \rangle d\tau w_{i,t} \mid \mathcal{F}_t \right] \right\| = \mathcal{O}(\varepsilon_t) \tag{A.14}$$

Now, for the second bound, since $u_i$ is continuous on a compact domain, it is bounded, and we readily get that:

$$\|\hat{v}_{i,t}\|_* = \mathcal{O}(1/\varepsilon_t) \tag{A.15}$$

∎

Moving forward, we provide some useful results from probability theory. The first two statements below are adapted from the classical textbook of Hall & Heyde [21], while the third one is a simplified version of [30, Theorem 3.2] on the recurrence of a nonnegative Markov process with diminishing drift. Namely, we have:

**Theorem A.1.** (Doob's maximal inequality, [21, Corollary 2.1]) *If $S_t$ is a martingale, we have:*

$$\mathbb{P}\left( \sup_{s \leq t} |S_s| > t \right) \leq \frac{\mathbb{E}[|S_t|]}{t} \quad \text{for all } t > 0. \tag{A.16}$$

**Theorem A.2.** (Burkholder's inequality, [21, Theorem 2.10]) *Let $S_t := \sum_{s=1}^t D_s$, where $(D_s)_{s \in \mathbb{N}}$ is a martingale difference sequence, and let $q \in (1, \infty)$. Then, there exists a constant $C$ that depends only on $q$ such that:*

$$\mathbb{E}[|S_t|^q] \leq C \, \mathbb{E}\left[ \left| \sum_{s=1}^t D_s^2 \right|^{q/2} \right] \tag{A.17}$$

**Theorem A.3.** (Lamperti [30, Theorem 3.2]) *Let the non-negative stochastic process $(x_t)_{t \in \mathbb{N}}$ be defined as*

$$x_{t+1} = (x_t + f(x_t) + \xi_t)^+ \tag{A.18}$$

*for some $x \mapsto f(x)$ bounded measurable function, and $\xi_t$ i.i.d. with $\mathbb{E}[\xi_t] = 0$, $\mathbb{V}(\xi_t) = \sigma^2 \neq 0$ and finite $2 + \varepsilon$ moment for some $\varepsilon > 0$. Then:*

*(i) if $f(x) \leq \sigma^2 / 2x$ for all $x$ large enough, the process is recurrent in the sense there exists $c < \infty$ such that*

$$\mathbb{P}\left( \liminf_{t \to \infty} x_t \leq c \right) = 1 \tag{A.19}$$

*(ii) if $f(x) \geq \theta \sigma^2 / 2x$ for some $\theta > 1$ and all $x$ large enough, the process is transient in the sense that*

$$\mathbb{P}\left( \lim_{t \to \infty} x_t = \infty \right) = 1 \tag{A.20}$$

# B  Analysis and results for strategic robustness

Our aim in this appendix is to provide a detailed proof for Theorem 1, which we restate below for convenience.

**Theorem 1.** *Let $x^* \in \mathcal{X}$ be a joint action profile in $\mathcal{G}(\mathcal{N}, \mathcal{X}, u)$. Then the following are equivalent:*

*(i) $x^*$ is a strategically robust equilibrium.*

*(ii) $\langle v(x^*), z \rangle \leq -m\|z\|$ for some $m > 0$ and all $z \in \mathrm{TC}(x^*)$, where $\mathrm{TC}(x^*)$ is the closure of all rays emanating from $x^*$ and intersecting $\mathcal{X}$ in at least one other point.*

*(iii) $v(x^*) \in \mathrm{int}(\mathrm{PC}(x^*))$, where $\mathrm{PC}(x^*) := \{y \in \mathcal{Y} : \langle y, z \rangle \leq 0, \text{ for all } z \in \mathrm{TC}(x^*)\}$.*

*Proof.* We will go full-circle by showing *(i)* $\implies$ *(ii)* $\implies$ *(iii)* $\implies$ *(i)*.

**($i$) $\implies$ ($ii$).** Suppose that $x^* \in \mathcal{X}$ is a strategically robust equilibrium, and let $\varepsilon > 0$ be such that $x^*$ is an equilibrium of any $\tilde{\mathcal{G}}$ with $\text{dist}(\mathcal{G}, \tilde{\mathcal{G}}) \leq \varepsilon$.

For the sake of contradiction, suppose that there exists $z \neq 0$, $z \in \text{TC}(x^*)$ such that

$$\langle v(x^*), z \rangle = 0. \tag{B.1}$$

which readily implies that there exists player $i \in \mathcal{N}$ and $z_i \neq 0$, $z_i \in \text{TC}_i(x_i^*)$, such that

$$\langle v_i(x^*), z_i \rangle = 0. \tag{B.2}$$

Fix some $y_i \in \mathcal{Y}_i$ such that $\langle y_i, z_i \rangle > 0$, and let $y \equiv (y_1, \ldots, y_N) \in \mathcal{Y}$ with $y_j \equiv 0$ for $j \neq i$, $j \in \mathcal{N}$. Using (B.1) and the definition of $y$, we get that $\langle v(x^*) + \varepsilon \|y\|_*^{-1} y, z \rangle > 0$, and therefore, there exists $p \in \mathcal{X}$ such that:

$$\langle v(x^*) + \varepsilon \|y\|_*^{-1} y, p - x^* \rangle > 0 \tag{B.3}$$

Now, define the game $\tilde{\mathcal{G}}$ with payoff functions

$$\tilde{u}_i(x) := u_i(x) + \varepsilon \|y\|_*^{-1} \langle y_i, x_i - x_i^* \rangle \tag{B.4}$$

and $\tilde{u}_j \equiv u_j$ for all $j \in \mathcal{N}$, $j \neq i$. Then, the individual gradient vector of player $i \in \mathcal{N}$ is given by

$$\tilde{v}_i(x) = v_i(x) + \varepsilon \|y\|_*^{-1} y_i \tag{B.5}$$

and the distance between $\mathcal{G}$ and $\tilde{\mathcal{G}}$ is equal to

$$\text{dist}(\mathcal{G}, \tilde{\mathcal{G}}) = \sup_{x \in \mathcal{X}} \|v(x) - \tilde{v}(x)\|_* = \varepsilon \|y\|_*^{-1} \|y\|_* = \varepsilon. \tag{B.6}$$

Finally, we conclude that $x^* \in \mathcal{X}$ is not an equilibrium of $\tilde{\mathcal{G}}$, since for $p \in \mathcal{X}$ as above, we have

$$\langle \tilde{v}(x^*), p - x^* \rangle = \langle v(x^*) + \varepsilon \|y\|_*^{-1} y, p - x^* \rangle > 0 \tag{B.7}$$

where the last inequality holds by (B.3). Thus, we arrive at a contradiction, i.e., $\langle v(x^*), z \rangle < 0$ for all $z \neq 0$, $z \in \text{TC}(x^*)$. Finally, since $\{z \in \mathcal{V} : z \in \text{TC}(x^*), \|z\| = 1\}$ is compact, we readily obtain

$$\sup\{\langle v(x^*), z \rangle : z \in \text{TC}(x^*), \|z\| = 1\} \leq -m \tag{B.8}$$

for some $m > 0$. Therefore, for all $z \in \text{TC}(x^*)$, we have:

$$\langle v(x^*), z \rangle \leq -m\|z\| \tag{B.9}$$

as was to be shown.

**($ii$) $\implies$ ($iii$).** First, note that the $\|\cdot\|_* -$ ball of radius $\varepsilon > 0$ centered at $v(x^*)$ can be written as:

$$\mathbb{B}_\varepsilon(v(x^*)) = v(x^*) + \varepsilon \, \mathbb{B}_1(0) \tag{B.10}$$

where $\mathbb{B}_\varepsilon(y) := \{y' \in \mathcal{Y} : \|y' - y\|_* \leq r\}$ for $y \in \mathcal{Y}$. Now, take any $y \in \mathbb{B}_1(0)$ and $z \in \text{TC}(x^*)$. Then, for $\varepsilon > 0$ we have

$$\begin{aligned}
\langle v(x^*) + \varepsilon y, z \rangle &= \langle v(x^*), z \rangle + \varepsilon \langle y, z \rangle \\
&\leq -m\|z\| + \varepsilon \|y\|_* \|z\| \\
&\leq -(m - \varepsilon)\|z\|
\end{aligned} \tag{B.11}$$

Setting $\varepsilon = m/2$, we have for all $z \in \text{TC}(x^*)$

$$\langle v(x^*) + (m/2)y, z \rangle < -(m/2)\|z\| \tag{B.12}$$

which implies that $v(x^*) + (m/2)y \in \text{PC}(x^*)$. Thus, we readily get that $\mathbb{B}_{m/2}(v(x^*)) \subseteq \text{PC}(x^*)$, i.e., $v(x^*) \in \text{int}(\text{PC}(x^*))$.

**($iii$) $\implies$ ($i$).** Suppose that $v(x^*) \in \text{int}(\text{PC}(x^*))$. First, it directly implies that $\langle v(x^*), z \rangle \leq 0$, for all $z \in \text{TC}(x^*)$, i.e., $x^*$ is an equilibrium of $\mathcal{G}$. In addition, there exists $\varepsilon > 0$ such that $\mathbb{B}_\varepsilon(v(x^*)) \subseteq \text{PC}(x^*)$. Therefore, for any game $\tilde{\mathcal{G}}$ with $\text{dist}(\mathcal{G}, \tilde{\mathcal{G}}) < \varepsilon$, we immediately get that $\tilde{v}(x^*) \in \text{PC}(x^*)$, which implies that $x^* \in \mathcal{X}$ is an equilibrium of $\tilde{\mathcal{G}}$. Thus, $x^*$ is strategically robust, and our proof is complete. $\blacksquare$

# C  Analysis and results for dynamic robustness

In this appendix, we provide detailed proofs of the statements presented in Section 4, along with several intermediate results that will serve as key building blocks.

**C.1. Intermediate results.** We begin this section with two results establishing sufficient conditions for convergence, followed by a high-probability deviation bound for martingales. We conclude with a variant of Farkas' Lemma, which will be instrumental in deriving convergence rates.

**Proposition C.1.** *Let $x^* \in \mathcal{X}$ and $\mathcal{Z} := \{z_1, \ldots, z_m\} \subseteq \mathcal{V}$ be a set of unit vectors, such that any $z \in \mathrm{TC}(x^*)$ can be written as $z = \sum_{j=1}^m \lambda_j z_j$ for some $\lambda_j \geq 0$. If $\lim_{t \to \infty} \langle y_t, z_j \rangle = -\infty$ for all $z_j \in \mathcal{Z}$, then $\lim_{t \to \infty} Q(y_t) = x^*$.*

*Proof.* Denote $Q(y_t)$ by $x_t$, and suppose that $\limsup_{t \to \infty} \|x_t - x^*\| > 0$. Then, there exists a subsequence $(x_{t_s})_{s \in \mathbb{N}}$ such that $\|x_{t_s} - x^*\|$ stays bounded away from zero, i.e., $\|x_{t_s} - x^*\| \geq c$ for some $c > 0$ and all $s \in \mathbb{N}$. Since $y_{t_s} \in \partial h(x_{t_s})$, we readily get for $z_{t_s} = (x_{t_s} - x^*)/\|x_{t_s} - x^*\|$:

$$
\begin{aligned}
h(x^*) &\geq h(x_{t_s}) + \langle y_{t_s}, x^* - x_{t_s} \rangle \\
&= h(x_{t_s}) - \langle y_{t_s}, z_{t_s} \rangle \|x_{t_s} - x^*\| \\
&\geq \min h - \langle y_{t_s}, z_{t_s} \rangle \|x_{t_s} - x^*\| .
\end{aligned}
\tag{C.1}
$$

Now, we have $z_{t_k} \in \mathrm{TC}(x^*)$, and by assumption, $z_{t_s} = \sum_{j=1}^m \lambda_{j,s} z_j$ for some coefficients $\lambda_{j,s} \geq 0$. Therefore, the above inequality can be written as:

$$
h(x^*) \geq \min h - \|x_{t_s} - x^*\| \sum_{j=1}^m \lambda_{j,s} \langle y_{t_s}, z_j \rangle
\tag{C.2}
$$

$$
\geq \min h - \left( \max_{j'} \langle y_{t_s}, z_{j'} \rangle \right) \|x_{t_s} - x^*\| \sum_{j=1}^m \lambda_{j,s}
\tag{C.3}
$$

Now, note that by the definition of $z_{t_s}$, we have $\|z_{t_s}\| = 1$, and, thus:

$$
1 = \|z_{t_s}\| = \left\| \sum_{j=1}^m \lambda_{j,s} z_j \right\| \leq \sum_{j=1}^m \lambda_{j,s} \|z_j\| = \sum_{j=1}^m \lambda_{j,s}
\tag{C.4}
$$

where we used that $\|z_j\| = 1$ for all $j$. Now, since $\lim_{t \to \infty} \langle y_t, z_j \rangle = -\infty$ for all $z_j \in \mathcal{Z}$, it readily implies that

$$
\lim_{t \to \infty} \max_{j'} \langle y_t, z_{j'} \rangle = -\infty
\tag{C.5}
$$

Therefore, for all $s$ large enough, we have $-\max_{j'} \langle y_{t_s}, z_{j'} \rangle > 0$, and, using that $\sum_{j=1}^m \lambda_{j,s} \geq 1$ and $\|x_{t_s} - x^*\| \geq c$ for all $s \in \mathbb{N}$, we obtain:

$$
h(x^*) \geq \min h + \left( -\max_{j'} \langle y_{t_s}, z_{j'} \rangle \right) \|x_{t_s} - x^*\| \sum_{j=1}^m \lambda_{j,s}
\tag{C.6}
$$

$$
\geq \min h - c \max_{j'} \langle y_{t_s}, z_{j'} \rangle
\tag{C.7}
$$

Finally, letting $s \to \infty$, we get that $h(x^*) \geq \infty$, which is a contradiction. Thus, the result follows. ∎

Finally, using the above proposition, we establish the following corollary.

**Corollary C.1.** *Let $\mathcal{W}(M) := \{y \in \mathcal{Y} : \max_{z \in \mathcal{Z}} \langle y, z \rangle < -M\}$ for $M > 0$. Then, for any $\varepsilon > 0$, there exists $M_\varepsilon > 0$ such that for all $y \in \mathcal{W}(M_\varepsilon)$ it holds $\|x^* - Q(y)\| < \varepsilon$.*

*Proof.* Suppose it does not hold. Then, there exists $\varepsilon > 0$ such that for any $t \in \mathbb{N}$, one can find $y_t \in \mathcal{Y}$ such that $\max_{z \in \mathcal{Z}} \langle y_t, z \rangle < -t$ and $\|x^* - Q(y_t)\| \geq \varepsilon$. Taking $t \to \infty$ leads to a contradiction with Proposition C.1. ∎

**Lemma C.1.** *Let $S_t := \gamma \sum_{s=1}^t \xi_s$ be a martingale with respect to a filtration $(\mathcal{F}_t)_{t \in \mathbb{N}}$ such that $\mathbb{E}[|\xi_t|^q] \leq \sigma_t^q$ for all $t \in \mathbb{N}$ and some $q \geq 2$. Then, for any $\mu \in (0, 1)$ and $c > 0$, it holds:*

$$\mathbb{P}\left(\sup_{\tau \leq t} |S_\tau| > c(\gamma t)^\mu\right) \leq C_q \frac{\gamma^{q(1-\mu)} \sum_{s=1}^t \sigma_s^q}{t^{1+q(\mu-1/2)}} \tag{C.8}$$

*where $C_q$ is a constant that depends only on $c$ and $q$.*

*Proof.* To bound the maximum absolute deviation of $S_t$, we apply Doob's maximal inequality (see Theorem A.1), and obtain:

$$\mathbb{P}\left(\sup_{\tau \leq t} |S_\tau| > c(\gamma t)^\mu\right) \leq \frac{\mathbb{E}[|S_t|^q]}{c^q (\gamma t)^{q\mu}} \tag{C.9}$$

Now, we invoke Burkholder's inequality (see Theorem A.2), from which we get:

$$\mathbb{E}[|S_t|^q] \leq C_q' \, \mathbb{E}\left[\left(\sum_{s=1}^t \gamma^2 |\xi_s|^2\right)^{q/2}\right] \leq C_q' \gamma^q \, \mathbb{E}\left[\left(\sum_{s=1}^t |\xi_s|^2\right)^{q/2}\right] \tag{C.10}$$

where $C_q$ is a constant that depends only on $q$. Since $q \geq 2$, applying Jensen's inequality, we obtain:

$$\left(\frac{1}{t} \sum_{s=1}^t |\xi_s|^2\right)^{q/2} \leq \frac{1}{t}\left(\sum_{s=1}^t |\xi_s|^q\right) \tag{C.11}$$

and, therefore,

$$\mathbb{E}\left[\left(\sum_{s=1}^t |\xi_s|^2\right)^{q/2}\right] \leq t^{q/2-1} \, \mathbb{E}\left[\sum_{s=1}^t |\xi_s|^q\right] \tag{C.12}$$

$$\leq t^{q/2-1} \sum_{s=1}^t \sigma_s^q \tag{C.13}$$

Thus, combining the above with (C.9) and (C.10), we obtain:

$$\mathbb{P}\left(\sup_{\tau \leq t} |S_\tau| > c(\gamma t)^\mu\right) \leq \frac{C_q' \gamma^q t^{q/2-1} \sum_{s=1}^t \sigma_s^q}{c^q (\gamma t)^{q\mu}} \tag{C.14}$$

$$\leq C_q \frac{\gamma^{q(1-\mu)} \sum_{s=1}^t \sigma_s^q}{t^{1+q(\mu-1/2)}} \tag{C.15}$$

for $C_q \equiv C_q'/c^q$, and the proof is complete. $\blacksquare$

We finally provide a separation result in the spirit of Farkas' lemma, that we will need for establishing the convergence rates.

**Lemma C.2.** *Let $\mathcal{X} = \{x \in \mathcal{V} : Ax = b, x \geq 0\}$ for $A \in \mathbb{R}^{m \times d}$, $b \in \mathbb{R}^d$. Then, for all $x^* \in \mathcal{X}$ with $act(x^*) := \{\beta \in \{1, \dots, d\} : x_\beta^* = 0\}$, there exists $P \equiv P(x^*) \geq 1$ such that for all $\mathcal{I} \subseteq act(x^*)$ at least one of the following is true:*

*(i) $\mathcal{I} \neq \emptyset$ and there exists $\beta \in act(x^*) \setminus \mathcal{I}$ such that $x_\beta \leq P \max\{x_\alpha : \alpha \in \mathcal{I}\}$ for all $x \in \mathcal{X}$.*

*(ii) There exists $z \in \ker(A)$ such that $\|z\| \leq P$, $z_\beta = 0$ for $\beta \in \mathcal{I}$ and $1 \leq z_\beta \leq P$ for $\beta \in act(x^*) \setminus \mathcal{I}$. Then, there exists*

*Proof.* For the proof, see Azizian et al. [6, Lemma 6]. $\blacksquare$

**C.2. Main results of Section 4.** With the necessary tools in place, we proceed to prove the main results stated in Section 4. We start with the first result, establishing that a non-equilibrium point cannot arise as a limit point of the sequence of play induced by (FTRL).

**Proposition 2.** *Suppose that* (FTRL) *is run with perfect gradient feedback of the form* $\hat{v}_t = v(x_t)$ *for all* $t = 1, 2, \ldots$, *and assume that* $x_t$ *converges to some* $\hat{x} \in \mathcal{X}$. *Then* $\hat{x}$ *is an equilibrium of* $\mathcal{G}$.

*Proof.* Since $\hat{x}$ is not an equilibrium, there exists $p \in \mathcal{X}$ with $\langle v(\hat{x}), p - \hat{x} \rangle > 0$. Therefore, by continuity of the function $x \mapsto \langle v(x), p - x \rangle$, there exists a neighborhood $\mathcal{U}$ of $\hat{x}$ and $c > 0$ such that $\langle v(x), p - x \rangle \geq c$ for all $x \in \mathcal{U}$.

Moreover, since $\mathrm{cl}(\mathcal{U})$ compact, we have $\sup_{x \in \mathrm{cl}(\mathcal{U})} \|v(x)\|_* = B < \infty$. For the sake of contradiction, suppose that $x_t \to \hat{x}$. Then, $x_t \in \mathcal{U} \cap \mathbb{B}_{c/4B}(\hat{x})$ eventually, i.e., there exists $n_0$ such that $x_t \in \mathcal{U}$ and $\|x_t - \hat{x}\| < c/4B$ for all $t \geq n_0$.

Finally, since $y_t \in \partial h(x_t)$, we have for $t > t_0$:

$$h(p) \geq h(x_t) + \langle y_t, p - x_t \rangle$$

$$\geq h(x_t) + \langle y_{t_0}, p - x_t \rangle + \gamma \sum_{s=t_0}^{t-1} \langle v(x_s), p - x_t \rangle$$

$$\geq h(x_t) + \langle y_{t_0}, p - x_t \rangle + \gamma \sum_{s=t_0}^{t-1} \langle v(x_s), p - x_s + x_s - x_t \rangle$$

$$\geq h(x_t) + \langle y_{t_0}, p - x_t \rangle + \gamma \sum_{s=t_0}^{t-1} (\langle v(x_s), p - x_s \rangle + \langle v(x_s), x_s - x_t \rangle)$$

$$\geq h(x_t) + \langle y_{t_0}, p - x_t \rangle + \gamma \sum_{s=t_0}^{t-1} (\langle v(x_s), p - x_s \rangle - \|v(x_s)\|_* \|x_s - x_t\|)$$

$$\geq h(x_t) + \langle y_{t_0}, p - x_t \rangle + \gamma \sum_{s=t_0}^{t-1} (c - B\|x_s - x_t\|)$$

$$\geq h(x_t) - \|y_{t_0}\|_* \|p - x_t\| + \gamma \sum_{s=t_0}^{t-1} (c - c/2)$$

$$\geq \min h - \|y_{t_0}\|_* \mathrm{diam}(\mathcal{X}) + \gamma c(t - t_0)/2 \tag{C.16}$$

Taking $t \to \infty$, we obtain $h(p) \geq \infty$, which is a contradiction. Therefore, the result follows. ∎

Moving forward, we show that equilibrium points in the relative interior cannot be limit points of (FTRL), either. Formally, we have:

**Proposition 3.** *Let* $x^* \in \mathrm{ri}(\mathcal{X})$ *be a Nash equilibrium of* $\mathcal{G}(\mathcal{N}, \mathcal{X}, u)$, *and* $(x_t)_{t \in \mathbb{N}}$ *be the sequence of play induced by* (FTRL) *with* $\hat{v}_t = v(x_t) + U_t$, *where* $U_t$ *i.i.d. with* $\mathbb{E}[U_t] = 0$ *and* $\mathrm{cov}(U_t) \succ 0$ *for all* $t \in \mathbb{N}$. *Then:*

$$\mathbb{P}\left( \lim_{t \to \infty} x_t = x^* \right) = 0 \qquad \text{for any } x_1 \in \mathcal{X}_h. \tag{15}$$

*Proof.* Since $x^*$ an equilibrium point in $\mathrm{ri}(\mathcal{X})$, we readily get that $\langle v(x^*), x - x^* \rangle = 0$ for all $x \in \mathcal{X}$, and $x^* \in \mathcal{X}_h$. In view of this, there exists $y^* \in \mathcal{Y}$ such that $y^* \in \partial h(x^*)$, i.e., $x^* = Q(y^*)$. Our goal is to show that the auxiliary process $y_t$ does not converge to $\partial h(x^*)$. However, there are infinitely many points in $\mathcal{Y}$ that belong to $\partial h(x^*)$, so this attempt is insufficient, in the sense that, showing that $y^*$ is not a limit point of the $y_t$ dynamics, does not preclude that some other $y \in \partial h(x^*)$ is not.

To tackle this issue, we will show that the space $\mathcal{Y}$ can be decomposed as $\mathcal{Y} = \widehat{\mathcal{Y}} \oplus \overline{\mathcal{Y}}$ where all the "essential" deviations of the problem is in $\overline{\mathcal{Y}}$. For this, we define the set

$$\widehat{\mathcal{Y}} = \{ y \in \mathcal{Y} : \langle y, p - x \rangle = 0, \text{ for all } x, p \in \mathcal{X} \}. \tag{C.17}$$

which is a subspace of $\mathcal{Y}$, and as the following lemma suggests, is equal to the polar cone at any point in the relative interior.

**Lemma C.3.** *Let* $x_0 \in \mathrm{ri}(\mathcal{X})$ *and* $\widehat{\mathcal{Y}} = \{ y \in \mathcal{Y} : \langle y, p - x \rangle = 0, \text{ for all } x, p \in \mathcal{X} \}$. *Then* $\widehat{\mathcal{Y}} = \mathrm{PC}(x_0)$.

To preserve the clarity of the argument, we defer the proof of Lemma C.3 until the end of this proposition. Letting $\overline{\mathcal{Y}}$ be the orthocomplement of $\widehat{\mathcal{Y}}$, we readily get that $\mathcal{Y} = \widehat{\mathcal{Y}} \oplus \overline{\mathcal{Y}}$, and any point $y$ in $\mathcal{Y}$ can be uniquely written as $y = \hat{y} + \bar{y}$ with $\hat{y} \in \widehat{\mathcal{Y}}$ and $\bar{y} \in \overline{\mathcal{Y}}$. Defining the linear map $\Pi : \mathcal{Y} \to \mathcal{Y}$ as $\Pi y = \bar{y}$, and more importantly, under all points in $\partial h(x^*)$ under $\Pi$ are essentially unique.

This is formalized in the following lemma, whose proof is relegated after this proposition.

**Lemma C.4.** *Let $x_0 \in \mathrm{ri}(\mathcal{X})$ and $y, y' \in \partial h(x_0)$. Then $\Pi y \in \partial h(x_0)$, and $\Pi y = \Pi y'$.*

In view of the above, we are now ready to prove the result. Namely, fix some $p \in \mathcal{X}$, $p \neq x^*$ and let $\xi_t := \langle \Pi U_t, p - x^* \rangle$.

Then, setting $\sigma^2 \equiv (p - x^*)^\top \Sigma (p - x^*) > 0$, we have $\xi_t \sim (0, \sigma^2)$ i.i.d., and, so, there exists $\varepsilon, \delta > 0$ such that $\mathbb{P}(\xi_t > \varepsilon) = \delta$ for all $t \in \mathbb{N}$. Therefore, by the second Borel-Cantelli lemma [9], we get $\mathbb{P}(A) = 1$ for $A \equiv \{\xi_t > \varepsilon$ infinitely often$\}$. For the sake of contradiction, suppose that $\mathbb{P}(B) > 0$ for $B \equiv \{\lim_{t\to\infty} x_t = x^*\}$. Fix some $\omega \in B$. Then, for all $t$ large enough, we readily get that $x_t(\omega) \in \mathrm{ri}(\mathcal{X})$, and, denoting $z_t := \Pi y_t$ and $z^* := \Pi y^*$, we readily get that

$$\lim_{t\to\infty} z_t(\omega) = z^* \tag{C.18}$$

Thus, setting $\alpha_t \equiv \langle z_t - z^*, p - x^* \rangle$ we conclude by the above equality that $\lim_{t\to\infty} \alpha_t = 0$, and therefore it holds

$$\begin{aligned}
0 &= \lim_{t\to\infty} (\alpha_t - \alpha_{t-1}) \\
&= \lim_{t\to\infty} \langle \Pi \hat{v}_t, p - x^* \rangle \\
&= \lim_{t\to\infty} \langle \Pi v(x_t), p - x^* \rangle + \langle \Pi U_t, p - x^* \rangle \\
&= \langle \Pi v(x^*), p - x^* \rangle + \lim_{t\to\infty} \xi_t \\
&= \lim_{t\to\infty} \xi_t \tag{C.19}
\end{aligned}$$

Therefore, $\omega \notin A$, which implies that $B \subseteq A^c$, with $\mathbb{P}(A^c) = 0$. Thus, $\mathbb{P}(B) = 0$, which is a contradiction, and the result follows.

∎

We now prove the two auxiliary lemmas presented in the proof of Proposition 3.

**Lemma C.3.** *Let $x_0 \in \mathrm{ri}(\mathcal{X})$ and $\widehat{\mathcal{Y}} = \{y \in \mathcal{Y} : \langle y, p - x \rangle = 0,$ for all $x, p \in \mathcal{X}\}$. Then $\widehat{\mathcal{Y}} = \mathrm{PC}(x_0)$.*

*Proof.* First, we will show that

$$\mathrm{PC}(x_0) = \{y \in \mathcal{Y} : \langle y, p - x_0 \rangle = 0 \text{ for all } p \in \mathcal{X}\} \tag{C.20}$$

For this, suppose that there exist $y \in \mathrm{PC}(x_0)$ and $p' \in \mathcal{X}$ such $\langle y, p' - x_0 \rangle < 0$. Then, since $x_0 \in \mathrm{ri}(\mathcal{X})$, there exists $\alpha > 0$ such that $x_0 - \alpha(p' - x_0) \in \mathcal{X}$. By the definition of the polar cone, $\langle y, x_0 - \alpha(p' - x_0) - x_0 \rangle \leq 0$, or equivalently, $\langle y, p' - x_0 \rangle \geq 0$, which is a contradiction. Therefore, (C.20) holds, which implies that $\widehat{\mathcal{Y}} \subseteq \mathrm{PC}(x)$.

Now, for the inverse inclusion, let $y \in \mathrm{PC}(x_0)$ and $p, x \in \mathcal{X}$. Then, we have:

$$\begin{aligned}
\langle y, p - x \rangle &= \langle y, p - x_0 + x_0 - x \rangle \\
&= \langle y, p - x_0 \rangle + \langle y, x_0 - x \rangle \\
&= 0 \tag{C.21}
\end{aligned}$$

where the last equality follows by (C.20). Thus, $y \in \widehat{\mathcal{Y}}$, and we conclude the result. ∎

**Lemma C.4.** *Let $x_0 \in \mathrm{ri}(\mathcal{X})$ and $y, y' \in \partial h(x_0)$. Then $\Pi y \in \partial h(x_0)$, and $\Pi y = \Pi y'$.*

*Proof.* For the first part, note that

$$\langle y, p - x_0 \rangle = \langle \hat{y} + \bar{y}, p - x_0 \rangle = \langle \hat{y}, p - x_0 \rangle + \langle \bar{y}, p - x_0 \rangle$$

$$= \langle \bar{y}, p - x_0 \rangle$$
$$= \langle \Pi y, p - x_0 \rangle \tag{C.22}$$

which directly implies that $\Pi y \in \partial h(x_0)$. For the second part, since $x_0 \in \mathrm{ri}(\mathcal{X})$, and $y, y' \in \partial h(x_0)$, we have that

$$\langle y - y', p - x_0 \rangle = 0 \quad \text{for all } p \in \mathcal{X} \tag{C.23}$$

Thus $y - y' \in \mathrm{PC}(x_0)$, and invoking Lemma C.3 we obtain that $y - y' \in \widehat{\mathcal{Y}}$. Therefore, applying the linear projection operator $\Pi$, we readily get that $\Pi(y - y') = 0$, and, using linearity, the result follows. ∎

We now turn to our main convergence theorems, showing that the iterates of (FTRL) converge with high probability under both gradient-based and payoff-based feedback

**Theorem 2.** *Let $x^* \in \mathcal{X}$ be a strategically robust equilibrium of $\mathcal{G}(\mathcal{N}, \mathcal{X}, u)$. Fix a confidence level $\delta > 0$, and let $(x_t)_{t \in \mathbb{N}}$ be the iterates of (FTRL) with feedback provided by (SFO), and step-size $\gamma > 0$ sufficiently small. Then, there exists a neighborhood $\mathcal{U}$ of $x^*$ in $\mathcal{X}_h$ such that:*

$$\mathbb{P}\left( \lim_{t \to \infty} x_t = x^* \right) \geq 1 - \delta \quad \text{if } x_1 \in \mathcal{U}. \tag{16}$$

*Proof.* Since $x^*$ strategically robust, $v(x^*)$ lies in the interior of the $\mathrm{PC}(x^*)$. By Lemma A.1, this implies in turn that there exists a polyhedral cone $\mathcal{K}$ generated by $\mathcal{Z} \equiv \{z_1, \ldots, z_r\}$ for $r \in \mathbb{N}$, such that $\mathrm{TC}(x^*) \subseteq \mathcal{K}$ and $\langle v(x^*), z \rangle < 0$ for all $z \in \mathcal{Z}$.[4] Therefore, for all $z \in \mathcal{Z}$, we have $\langle v(x^*), z \rangle \leq -m$, and by continuity of the vector field $v$, there exists a neighborhood $\mathcal{U}$ of $x^*$ and $c > 0$ such that $\langle v(x), z \rangle \leq -c$ for all $z \in \mathcal{Z}$ and $x \in \mathcal{U}$.

Fixing some $z \in \mathcal{Z}$, we obtain:

$$\begin{aligned} \langle y_{t+1}, z \rangle &= \langle y_t, z \rangle + \gamma \langle \hat{v}_t, z \rangle \\ &= \langle y_t, z \rangle + \gamma \langle v(x_t), z \rangle + \gamma \langle U_t, z \rangle \\ &= \langle y_1, z \rangle + \gamma \sum_{s=1}^{t} \langle v(x_s), z \rangle + \gamma \sum_{s=1}^{t} \langle U_s, z \rangle \end{aligned} \tag{C.24}$$

Now, we define the stochastic process $(S_t)_{t \in \mathbb{N}}$ via $S_t := \gamma \sum_{s=1}^{t} \langle U_s, z \rangle$, which is a martingale, since $\mathbb{E}[\langle U_s, z \rangle \mid \mathcal{F}_s] = 0$.

Therefore, by Lemma C.1 for $\sigma_t \equiv \sigma$, $q > 2$ and $\mu \in (0, 1)$, whose value is determined later, we get:

$$\delta_t := \mathbb{P}\left( \sup_{\tau \leq t} |S_\tau| > c(\gamma t)^\mu \right) \leq C_q \frac{\gamma^{q(1-\mu)} \sigma^q}{t^{q(\mu - 1/2)}} \tag{C.25}$$

where $C_q$ is a constant that depends only on $c$ and $q$. Thus, we readily have that:

$$\begin{aligned} \mathbb{P}\left( \bigcap_{t \geq 1} \left\{ \sup_{\tau \leq t} |S_\tau| \leq c(\gamma t)^\mu \right\} \right) &= 1 - \mathbb{P}\left( \bigcup_{t \geq 1} \left\{ \sup_{\tau \leq t} |S_\tau| > c(\gamma t)^\mu \right\} \right) \\ &\geq 1 - \sum_{t=1}^{\infty} \mathbb{P}\left( \sup_{\tau \leq t} |S_\tau| > c(\gamma t)^\mu \right) \\ &\geq 1 - \sum_{t=1}^{\infty} \delta_t \end{aligned} \tag{C.26}$$

where the second inequality comes from the union bound. Now, we need to ensure that $\sum_{t=1}^{\infty} \delta_t \leq \delta/r$. For this, we need the sequence to be summable, which, using (C.25), is guaranteed for $q(\mu - 1/2) > 1$, or equivalently, $\mu \in (1/2 + 1/q, 1)$. Therefore, for $\gamma > 0$ small enough, we obtain that $\sum_{t=1}^{\infty} \delta_t \leq \delta/r$.

Therefore, with probability at least $1 - \delta/r$, the template inequality becomes:

$$\langle y_{t+1}, z \rangle \leq \langle y_1, z \rangle + \gamma \sum_{s=1}^{t} \langle v(x_s), z \rangle + c(\gamma t)^\mu \tag{C.27}$$

---

[4]To resolve any ambiguities, the cone in question here is the polar of the cone provided by Lemma A.1.

If we initialize $y_1$ such that $\langle y_1, z' \rangle < -M - c$ for all $z' \in \mathcal{Z}$, we get that $\langle y_t, z \rangle < -M$ for all $t \in \mathbb{N}$ with probability at least $1 - \delta/r$. To see this, suppose that $\langle y_s, z \rangle < -M$ for all $s = 1, \dots, t$. Then

$$\langle y_{t+1}, z \rangle = \langle y_1, z \rangle + \gamma \sum_{s=1}^{t} \langle v(x_s), z \rangle + \gamma \sum_{s=1}^{t} \langle U_s, z \rangle$$
$$\le -M - c - c\gamma t + c(\gamma t)^{\mu} \qquad (C.28)$$

For $t \in \mathbb{N}$ with $\gamma t < 1$, we have $-c + c(\gamma t)^{\mu} < 0$, while for $\gamma t \ge 1$, it holds $-c\gamma t + c(\gamma t)^{\mu} < 0$. In both cases, we conclude that $\langle y_{t+1}, z \rangle < -M$, and by induction, we get the inequality.

Therefore, with probability at least $1 - \delta/r$, we have:

$$\langle y_{t+1}, z \rangle \le -M - c - c\gamma t + c(\gamma t)^{\mu} \qquad (C.29)$$

and sending $t \to \infty$, we get $\langle y_t, z \rangle \to -\infty$.

Finally, repeating the same argument for all $z \in \mathcal{Z}$ and applying a union bound, we readily get that $\langle y_t, z \rangle \to -\infty$ with probability at least $1 - \delta$, and invoking [Proposition C.1](#), the result follows. ∎

Having established the local convergence to $x^*$ with high probability, we proceed to the convergence rate in the case of affinely constrained $\mathcal{X}$ and decomposable regularizer $h$.

**Theorem 3.** *If, in addition, $\mathcal{X}$ is a polyhedral domain and $h$ is decomposable with kernel $\theta$, on the event $E := \{\lim_{t \to \infty} x_t = x^*\}$ it holds:*

$$\|x_t - x^*\| = \phi(-\Theta(t)) \qquad (17)$$

*where $\phi$ is the rate function defined via*

$$\phi(z) := \begin{cases} (\theta')^{-1}(z) & \text{if } z > \theta'(0^+) \\ 0 & \text{if } z \le \theta'(0^+) \end{cases} \qquad (18)$$

*Proof.* By the definition of the iterates of ([FTRL](#)), we have:

$$Q(y_t) = \arg\min_{x \in \mathcal{X}}\{h(x) - \langle y_t, x \rangle : Ax = b, x \ge 0\} \qquad (C.30)$$

Introducing the Lagrangian

$$\mathcal{L}(x, \lambda, \mu) = h(x) - \langle y_t, x \rangle + \sum_{i=1}^{m} \lambda_i(a_i^\top x - b_i) - \sum_{j=1}^{d} \mu_j x_j \qquad (C.31)$$

with $\lambda_i \in \mathbb{R}$ and $\mu_j \ge 0$, by the KKT conditions, we readily obtain:

$$y_t = \nabla h(x_t) + \sum_{\tau=1}^{m} \lambda_i a_i - \mu \qquad (C.32)$$

where $\nabla h(x) = \sum_{\beta=1}^{d} \theta'(x_{\beta,t})e_\beta$, since $\theta$ is continuously differentiable.

For the sequel, we define the set of active constraints at $x^*$ as $\text{act}(x^*) := \{\beta \in \{1, \dots, d\} : x_\beta^* = 0\}$. Note that on the event of $\{\lim_{t \to \infty} x_t = x^*\}$, the iterates $x_t$ lie in a neighborhood of $x^*$, as shown in [Theorem 2](#). Thus, all non-active indices $\alpha \notin \text{act}(x^*)$ stay bounded away from zero, and so $|\theta(x_{\alpha,t})|$ remains bounded for all $t$.

We treat the two cases separately: (i) the steep case, where $h$ is steep – equivalently $\theta(0^+) = -\infty$, and (ii) the non-steep case, where is $h$ not steep, i.e., $\theta'(0^+) > -\infty$.

**The steep case.** We define the set of "good" indices $\mathcal{I}$ at step $t$ as: $\beta \in \mathcal{I}$ if $\theta'(x_{\beta,t}) \le -\Theta(t)$. Our goal is to show that all indices $\text{act}(x^*)$ of $x_t$ are "good". Fix some $t \in \mathbb{N}$.

Suppose that $\text{act}(x^*) \setminus \mathcal{I} \ne \emptyset$, and let $P \ge 1$, as per [Lemma C.2](#). Then,

- If condition *(i)* of *Lemma C.2* holds, there exists $\beta'$ such that $x_{\beta',t} \le P \max\{x_{\alpha,t} : \alpha \in \mathcal{I}\}$, and thus, $\mathcal{I} \leftarrow \mathcal{I} \cup \{\beta'\}$.

- If condition *(ii)* of *Lemma C.2* holds, there exists $z' \in \ker(A)$ such that $\|z'\| \le P$, $z'_\beta = 0$ if $\beta \in \mathcal{I}$ and $1 \le z'_\beta \le P$ if $\beta \in \mathrm{act}(x^*) \setminus \mathcal{I}$. By (C.32), and noting that $z' \in \ker(A)$ and $\mu = 0$, since all constraints are non-active due to steepness of $h$, we have:

$$\langle \nabla h(x_t), z' \rangle = \langle y_t, z' \rangle \tag{C.33}$$

Moreover, it holds:

$$\langle \nabla h(x_t), z' \rangle = \sum_{\beta=1}^{d} \theta'(x_{\beta,t}) z'_\beta = \sum_{\beta \in \mathcal{I}} \theta'(x_{\beta,t}) z'_\beta + \sum_{\beta \in \mathrm{act}(x^*) \setminus \mathcal{I}} \theta'(x_{\beta,t}) z'_\beta + \sum_{\beta \notin \mathrm{act}(x^*)} \theta'(x_{\beta,t}) z'_\beta$$
$$= \sum_{\beta \in \mathrm{act}(x^*) \setminus \mathcal{I}} \theta'(x_{\beta,t}) z'_\beta + C \tag{C.34}$$

for a constant $C$, since all non-active indices remain bounded away from zero, as explained in the beginning. Now, note that $z' \in \mathrm{TC}(x^*)$, and thus, by Lemma A.1, we can write $z'$ as $z' = \sum_{i=1}^{r} \ell_i z_i$ with $\ell_i \ge 0$, such that $\langle y_t, z_i \rangle \le -\Theta(t)$ for all $i = 1, \ldots, r$ as in the proof of Theorem 2. So, combining it with (C.39), (C.34), we obtain:

$$\sum_{\beta \in \mathrm{act}(x^*) \setminus \mathcal{I}} \theta'(x_{\beta,t}) z'_\beta \le -\Theta(t) \tag{C.35}$$

and therefore, there exists at least one $\beta' \in \mathrm{act}(x^*) \setminus \mathcal{I}$ such that

$$\theta'(x_{\beta',t}) z'_{\beta'} \le -\Theta(t) \tag{C.36}$$

Thus, $\mathcal{I} \leftarrow \mathcal{I} \cup \{\beta'\}$.

Therefore, as $\mathrm{act}(x^*)$ is finite, we conclude inductively that $\theta'(x_{\beta,t}) \le -\Theta(t)$ for all $\beta \in \mathrm{act}(x^*)$. Finally, we have that $\mathbb{R}^d = \mathrm{row}(A) + \mathrm{span}\{e_\beta : \beta \in \mathrm{act}(x^*)\}$, and thus, for all $i$, we can write the standard basis vector $e_i$ as $e_i = \sum_{\beta \in \mathrm{act}(x^*)} \lambda_{i,\beta} e_\beta + a_i$ for some $a_i \in \mathrm{row}(A)$

$$x_{i,t} - x_i^* = \langle x_t - x^*, e_i \rangle = \left\langle x_t - x^*, \sum_{\beta \in \mathrm{act}(x^*)} \lambda_{i,\beta} e_\beta + a_i \right\rangle$$
$$= \left\langle x_t - x^*, \sum_{\beta \in \mathrm{act}(x^*)} \lambda_{i,\beta} e_\beta \right\rangle$$
$$= \sum_{\beta \in \mathrm{act}(x^*)} \lambda_{i,\beta} x_{\beta,t} \tag{C.37}$$

where we used that $\langle x_t - x^*, a_i \rangle = 0$. Thus, since $\theta'(x_{\beta,t}) \le -\Theta(t)$ for all $\beta \in \mathrm{act}(x^*)$, by the equivalence of norms and the above, we conclude that

$$\|x_t - x^*\| = (\theta')^{-1}(-\Theta(t)) \tag{C.38}$$

**The non-steep case.** For the non-steep case, we follow a similar approach, but with some modifications since the iterates of (FTRL) are not always in the interior of $\mathcal{X}$.

Specifically, let the set of "good" indices $\mathcal{I}$ be defined as: $\beta \in \mathcal{I}$ if $x_{\beta,t} = 0$ or $\theta'(x_{\beta,t}) \le -\Theta(t)$. Our goal is to show that all indices $\mathrm{act}(x^*)$ of $x_t$ are "good". We construct $\mathcal{I}$ sequentially, as before.

Suppose that $\mathrm{act}(x^*) \setminus \mathcal{I} \ne \emptyset$, and let $P \ge 1$, as per Lemma C.2. Then,

- If condition *(i)* of *Lemma C.2* holds, there exists $\beta'$ such that $x_{\beta',t} \le P \max\{x_{\alpha,t} : \alpha \in \mathcal{I}\}$, and thus, $\mathcal{I} \leftarrow \mathcal{I} \cup \{\beta'\}$.

- If condition *(ii)* of *Lemma C.2* holds, there exists $z' \in \ker(A)$ such that $\|z'\| \le P$, $z'_\beta = 0$ if $\beta \in \mathcal{I}$ and $1 \le z'_\beta \le P$ if $\beta \in \mathrm{act}(x^*) \setminus \mathcal{I}$. Therefore, we have

$$\langle \nabla h(x_t), z' \rangle = \langle y_t, z' \rangle + \langle \mu, z' \rangle = \langle y_t, z' \rangle + \sum_{\beta \in \mathcal{I}} \mu_\beta z'_\beta + \sum_{\beta \in \mathrm{act}(x^*) \setminus \mathcal{I}} \mu_\beta z'_\beta + \sum_{\beta \notin \mathrm{act}(x^*)} \mu_\beta z'_\beta$$
$$= \langle y_t, z' \rangle \tag{C.39}$$

where, in this case, we used that (i) $z'_\beta = 0$ for $\beta \in \mathcal{I}$, (ii) $\mu_\beta = 0$ by complementary slackness for $\beta \notin \text{act}(x^*)$ since these constraints remain non-active for the whole process, and (iii) $\mu_\beta = 0$, again by complementary slackness for $\beta \in \text{act}(x^*) \setminus \mathcal{I}$ since if they were active, we would have $\beta \in \mathcal{I}$. This, with the same argument as before, we conclude that

$$\sum_{\beta \in \text{act}(x^*) \setminus \mathcal{I}} \theta'(x_{\beta,t}) z'_\beta \leq -\Theta(t) \tag{C.40}$$

and therefore, there exists at least one $\beta' \in \text{act}(x^*) \setminus \mathcal{I}$ such that

$$\theta'(x_{\beta',t}) z'_{\beta'} \leq -\Theta(t) \tag{C.41}$$

This holds until $\beta'$ vanishes, which can lead to $\mu_{\beta'} > 0$. In either case, we have $\mathcal{I} \leftarrow \mathcal{I} \cup \{\beta'\}$.

Finally, since $\text{act}(x^*)$ is finite, we conclude inductively that all for all $\beta \in \text{act}(x^*)$, we have either $\theta'(x_{\beta,t}) \leq -\Theta(t)$ or $x_{\beta,t} = 0$. As in the steep case, we conclude

$$\|x_t - x^*\| = \phi(-\Theta(t)) \tag{C.42}$$

■

We now shift to the payoff-based setting. The relevant result is restated below.

**Theorem 4.** *Let $x^* \in \mathcal{X}$ be a strategically robust equilibrium of $\mathcal{G}$. Fix a confidence level $\delta > 0$, and let $(x_t)_{t \in \mathbb{N}}$ be the iterates of* (FTRL) *run with* (SPSA) *with $\varepsilon_t \propto 1/t^p$ for some $p \in (0, 1/2)$ and step-size $\gamma > 0$ sufficiently small. Then, there exists a neighborhood $\mathcal{U}$ of $x^*$ such that:*

$$\mathbb{P}\left(\lim_{t \to \infty} x_t = x^*\right) \geq 1 - \delta \qquad \text{if } x_1 \in \mathcal{U}. \tag{19}$$

*If, in addition, $\mathcal{X}$ is affinely constrained and $h$ is decomposable with kernel $\theta$, then, whenever $x_t$ converges to $x^*$, we have:*

$$\|x_t - x^*\| = \phi(-\Theta(t)). \tag{20}$$

*Proof.* First of all, we write $\hat{v}_t$ in the following convenient form:

$$\hat{v}_t = v(x_t) + U_t + b_t \tag{C.43}$$

with

$$U_t = \hat{v}_t - \mathbb{E}[\hat{v}_t \mid \mathcal{F}_t] \qquad \text{and} \qquad b_t = \mathbb{E}[\hat{v}_t \mid \mathcal{F}_t] - v(x_t) \tag{C.44}$$

which, by Proposition A.1, satisfy the bounds $\|U_t\|_* = \mathcal{O}(1/\varepsilon_t)$ and $\|b_t\|_* = \mathcal{O}(\varepsilon_t)$. Now, as in the proof of Theorem 2, $v(x^*)$ lies in the interior of the PC$(x^*)$. By Lemma A.1 this in turn implies that there exists a polyhedral cone $\mathcal{K}$ generated by $\mathcal{Z} \equiv \{z_1, \ldots, z_r\}$ for $r \in \mathbb{N}$, such that TC$(x^*) \subseteq \mathcal{K}$ and $\langle v(x^*), z \rangle < 0$ for all $z \in \mathcal{Z}$. [5] Therefore, for all $z \in \mathcal{Z}$, we have $\langle v(x^*), z \rangle \leq -m$, and by continuity of the vector field $v$, there exists a neighborhood $\mathcal{U}$ of $x^*$ and $c > 0$ such that $\langle v(x), z \rangle \leq -c$ for all $z \in \mathcal{Z}$ and $x \in \mathcal{U}$. Fix some $z \in \mathcal{Z}$. Then, unfolding the evolution of $y_t$, we have:

$$\begin{aligned} \langle y_{t+1}, z \rangle &= \langle y_t, z \rangle + \gamma \langle \hat{v}_t, z \rangle \\ &= \langle y_t, z \rangle + \gamma \langle v(x_t), z \rangle + \gamma \langle U_t, z \rangle + \gamma \langle b_t, z \rangle \\ &= \langle y_1, z \rangle + \gamma \sum_{s=1}^{t} \langle v(x_s), z \rangle + \gamma \sum_{s=1}^{t} \langle U_s, z \rangle + \gamma \sum_{s=1}^{t} \langle b_s, z \rangle \end{aligned} \tag{C.45}$$

Now, we define the stochastic process $(S_t)_{t \in \mathbb{N}}$ via $S_t := \gamma \sum_{s=1}^{t} \langle U_s, z \rangle$, which is a martingale, since $\mathbb{E}[\langle U_s, z \rangle \mid \mathcal{F}_s] = 0$, and $\mathbb{E}[|\langle U_s, z \rangle|^q \mid \mathcal{F}_s] \leq \mathbb{E}[\|U_s\|_*^q \mid \mathcal{F}_s] = \mathcal{O}((1/\varepsilon_t)^q)$.

Therefore, by Lemma C.1 for $\sigma_t = \Theta(1/\varepsilon_t)$, $q > 2$ and $\mu \in (0, 1)$, whose value is determined later, we get:

$$\delta_t := \mathbb{P}\left(\sup_{\tau \leq t} |S_\tau| > (c/2)(\gamma t)^\mu\right) \leq C_q \frac{\gamma^{q(1-\mu)} \sum_{s=1}^{t} \sigma_s^q}{t^{1+q(\mu-1/2)}} \tag{C.46}$$

---

[5] As before, to resolve any ambiguities, the cone in question here is the polar of the cone provided by Lemma A.1.

where $C_q$ is a constant that depends only on $c$ and $q$. Thus, for $\varepsilon_t = \varepsilon/t^p$, there exist $B > 0$ such that:

$$\sum_{s=1}^{t} \sigma_s^q \leq B\varepsilon^{-q} \sum_{s=1}^{t} s^{pq} \leq B'\varepsilon^{-q} t^{1+pq} \tag{C.47}$$

where we used that $\sum_{s=1}^{t} s^{pq} = \Theta(t^{1+pq})$. So, using the above bound, (C.48) becomes:

$$\delta_t \leq C_q' \frac{\gamma^{q(1-\mu)}\varepsilon^{-q}t^{1+pq}}{t^{1+q(\mu-1/2)}} \leq C_q' \frac{\gamma^{q(1-\mu)}\varepsilon^{-q}}{t^{q(\mu-1/2-p)}} \tag{C.48}$$

Thus, we readily have that:

$$\mathbb{P}\left(\bigcap_{t\geq 1}\left\{\sup_{\tau\leq t}|S_\tau| \leq c(\gamma t)^\mu\right\}\right) = 1 - \mathbb{P}\left(\bigcup_{t\geq 1}\left\{\sup_{\tau\leq t}|S_\tau| > c(\gamma t)^\mu\right\}\right)$$

$$\geq 1 - \sum_{t=1}^{\infty} \mathbb{P}\left(\sup_{\tau\leq t}|S_\tau| > c(\gamma t)^\mu\right)$$

$$\geq 1 - \sum_{t=1}^{\infty} \delta_t \tag{C.49}$$

Now, we need to show that there exists $\mu \in (0, 1)$ and $q > 2$ such that $\sum_{t=1}^{\infty} \delta_t \leq \delta/r$. In order for the sum to be finite, we need $q(\mu - 1/2 - p) > 1$ which readily implies that $p < \mu - 1/2 - 1/q$.

For the bias term, since $\|b_s\|_* = \Theta(\varepsilon_s)$, we have:

$$\sum_{s=1}^{t}\langle b_s, z\rangle \leq \sum_{s=1}^{t}\|b_s\|_*\|z\| \leq \sum_{s=1}^{t}\|b_s\|_* \leq D\sum_{s=1}^{t}\varepsilon_s \leq D\sum_{s=1}^{t}\varepsilon/s^p \leq D'\varepsilon t^{1-p} \tag{C.50}$$

for some $D' > 0$, where in the last inequality we used that $\sum_{s=1}^{t} 1/s^p = \Theta(t^{1-p})$.

Therefore, for $1 - \mu < p$, and $\gamma < 1$, we readily get that

$$\gamma\sum_{s=1}^{t}\langle b_s, z\rangle \leq D'\gamma\varepsilon t^\mu \leq D'\varepsilon(\gamma t)^\mu \tag{C.51}$$

Therefore, we need to satisfy

$$1 - \mu < p < \mu - 1/2 - 1/q \tag{C.52}$$

from which we obtain $\mu \in (3/4, 1)$. Thus, for $p \in (0, 1/2)$, there exist $\mu \in (3/4, 1)$ and $q > 2$ that satisfy (C.51). So, for $\varepsilon, \gamma$ sufficiently small we can guarantee that

$$\gamma\sum_{s=1}^{t}\langle b_s, z\rangle \leq (c/2)(\gamma t)^\mu \qquad \text{and} \qquad \sum_{t=1}^{\infty}\delta_t \leq \delta/r \tag{C.53}$$

Therefore, with probability at least $1 - \delta/r$, the template inequality becomes:

$$\langle y_{t+1}, z\rangle \leq \langle y_1, z\rangle + \gamma\sum_{s=1}^{t}\langle v(x_s), z\rangle + (c/2)(\gamma t)^\mu + (c/2)(\gamma t)^\mu$$

$$\leq \langle y_1, z\rangle + \gamma\sum_{s=1}^{t}\langle v(x_s), z\rangle + c(\gamma t)^\mu \tag{C.54}$$

Initializing $y_1$ such that $\langle y_1, z'\rangle < -M - c$ for all $z' \in \mathcal{Z}$, we have $\langle y_t, z\rangle < -M$ for all $t \in \mathbb{N}$ with probability at least $1 - \delta/r$. To see this, we proceed by induction, and suppose that $\langle y_s, z\rangle < -M$ for all $s = 1, \ldots, t$. Then

$$\langle y_{t+1}, z\rangle = \langle y_1, z\rangle + \gamma\sum_{s=1}^{t}\langle v(x_s), z\rangle + \gamma\sum_{s=1}^{t}\langle U_s, z\rangle + \gamma\sum_{s=1}^{t}\langle b_s, z\rangle$$

$$\leq -M - c - c\gamma t + c(\gamma t)^\mu \tag{C.55}$$

For $t \in \mathbb{N}$ with $\gamma t < 1$, we have $-c + c(\gamma t)^\mu < 0$, while for $\gamma t \geq 1$, it holds $-c\gamma t + c(\gamma t)^\mu < 0$. In both cases, we conclude that $\langle y_{t+1}, z\rangle < -M$, and by induction, we get the inequality.

Therefore, with probability at least $1 - \delta/r$, we have:

$$\langle y_{t+1}, z \rangle \leq -M - c - c\gamma t + c(\gamma t)^\mu \tag{C.56}$$

and sending $t \to \infty$, we get $\langle y_t, z \rangle \to -\infty$.

As a final step, using the same argument for all $z \in \mathcal{Z}$ and applying a union bound, we have that $\langle y_t, z \rangle \to -\infty$ with probability at least $1 - \delta$, and by Proposition C.1, we get that

$$\mathbb{P}\left(\lim_{t\to\infty} x_t = x^*\right) \geq 1 - \delta . \tag{C.57}$$

If, in addition, $\mathcal{X}$ is affinely constrained and $h$ is decomposable with kernel $\theta$, then the argument in the proof of Theorem 3 applies verbatim, yielding:

$$\|x_t - x^*\| = \phi(-\Theta(t)) . \tag{C.58}$$

whenever $x_t$ converges to $x^*$. ∎

We conclude this appendix with Proposition 4, which illustrates that an extreme, non-strategically robust equilibrium may exhibit fundamentally different behavior depending on the choice of regularizer.

**Proposition 4.** *Consider the 1-player game $\mathcal{G}$ with $\mathcal{X} = [0,1]$, $u(x) = -\frac{3}{4}x^{4/3}$ and $x^* = 0$. Let $(x_t)_{t\in\mathbb{N}}$ be the iterates of (FTRL) with $\gamma < 1$, and $\hat{v}_t = v(x_t) + U_t$, where $U_t$ are i.i.d. standard normal random variables for all $t \in \mathbb{N}$. Then, for any initial condition $y_1 \in \mathbb{R}$, we have:*

*(i) For $h(x) = x \log x$, it holds $\mathbb{P}(\lim_{t\to\infty} x_t = x^*) = 0$.*

*(ii) For $h(x) = -2\sqrt{x}$, it holds $\mathbb{P}(\lim_{t\to\infty} x_t = x^*) = 1$.*

*Proof.* We show each case separately.

(i) Writing down the (FTRL) dynamics, we have

$$
\begin{aligned}
y_{t+1} &= y_t + \gamma(-x_t^{1/3} + U_t) \\
x_t &= \sup_{x\in[0,1]} (y_t x - x \log x)
\end{aligned}
\tag{C.59}
$$

Solving the maximization problem in the definition of $x_t$, we obtain:

$$
x_t = \begin{cases} \exp(y_t - 1), & \text{if } y_t \leq 1 \\ 1, & \text{if } y_t > 1 \end{cases}
$$

or, equivalently, $x_t = \mathbb{1}(y_t > 1) + \mathbb{1}(y_t \leq 1)\exp(y_t - 1)$ and the dual process can be written as:

$$y_{t+1} = y_t - \gamma \mathbb{1}(y_t > 1) - \gamma \mathbb{1}(y_t \leq 1)\exp((y_t - 1)/3) + \gamma U_t \tag{C.60}$$

It is clear that $x_t \to 0$ if and only if $y_t \to -\infty$ as $t$ goes to infinity. For notational convenience, set $z_n \equiv -y_t$. Then, the evolution of the dual process becomes:

$$z_{t+1} = z_t + \gamma \mathbb{1}(z_t < -1) + \gamma \mathbb{1}(z_t \geq -1)\exp((-z_t - 1)/3) - \gamma U_t \tag{C.61}$$

Now, define the process

$$z'_{t+1} \equiv \left(z'_t + \gamma \mathbb{1}(z'_t < -1) + \gamma \mathbb{1}(z'_t \geq -1)\exp((-z'_t - 1)/3) - \gamma U_t\right)^+, \quad z'_1 = z_1 \tag{C.62}$$

where $U_t$ is the same random variable as in (C.61).

The rest of our proof relies on a series of claims, which we state and prove one-by-one.

**Claim 1.** *The process $(z'_t)_{t\in\mathbb{N}}$ dominates $(z_t)_{t\in\mathbb{N}}$, i.e., $z'_t \geq z_t$ for all $t \in \mathbb{N}$.*

The proof of Claim 1 lies at the end. Now, invoking Theorem A.3 with

$$f(z) \equiv \gamma \mathbb{1}(z < -1) + \gamma \mathbb{1}(z \geq -1)\exp((-z - 1)/3) \tag{C.63}$$

bounded and $\sigma^2 = \gamma^2$, it holds that

$$f(z) \leq \frac{\sigma^2}{2z} \quad \text{for all } z \text{ large enough} \tag{C.64}$$

Thus, $(z'_t)_{t \in \mathbb{N}}$ is recurrent, which implies that

$$\mathbb{P}\left( \lim_{t \to \infty} z'_t = \infty \right) = 0 \tag{C.65}$$

Finally, since $(z'_t)_{t \in \mathbb{N}}$ dominates $(z_t)_{t \in \mathbb{N}}$ by Claim 1, we obtain

$$\left\{ \lim_{t \to \infty} z_t = \infty \right\} \subseteq \left\{ \lim_{t \to \infty} z'_t = \infty \right\} \tag{C.66}$$

which implies that

$$\mathbb{P}\left( \lim_{t \to \infty} x_t = x^* \right) = \mathbb{P}\left( \lim_{t \to \infty} z_t = \infty \right) \le \mathbb{P}\left( \lim_{t \to \infty} z'_t = \infty \right) = 0 \tag{C.67}$$

and the result follows.

*Proof of Claim 1.* Consider the function

$$g(z) := z + \gamma \, \mathbb{1}(z < -1) + \gamma \, \mathbb{1}(z \ge -1) \exp\left( (-z - 1)/3 \right) \tag{C.68}$$

Then:

- For $z < -1$: $g'(z) = 1$

- For $z > -1$: $g'(z) = 1 - \gamma/3 \exp\left( (-z - 1)/3 \right) > 1 - \gamma/3 > 0$

Thus, $g$ is strictly increasing for all $z \in \mathbb{R}$. Now, for the sake of contradiction, suppose that there exists $\omega \in \Omega$ and a first time $k + 1 \in \mathbb{N}$ where the dominance does not hold, i.e.,

$$z_k(\omega) \le z'_k(\omega) \quad \text{and} \quad z'_{k+1}(\omega) < z_{k+1}(\omega) \tag{C.69}$$

By the monotonicity property of $g$, we get that

$$g(z_k(\omega)) \le g(z'_k(\omega)) \tag{C.70}$$

and, therefore, adding $-\gamma U_s(\omega)$ in both sides

$$\begin{aligned} z_{s+1}(\omega) &\le z'_s + \gamma \, \mathbb{1}(z'_t < -1) + \gamma \, \mathbb{1}(z'_t \ge -1) \exp\left( (-z'_t - 1)/3 \right) - \gamma U_s \\ &\le \left( z'_s + \gamma \, \mathbb{1}(z'_t < -1) + \gamma \, \mathbb{1}(z'_t \ge -1) \exp\left( (-z'_t - 1)/3 \right) - \gamma U_s \right)^+ \\ &\le z'_{s+1}(\omega) \end{aligned} \tag{C.71}$$

which is a contradiction. Thus, the proof of Claim 1 is complete.

(ii) In this setup, the (FTRL) dynamics are described by the system

$$\begin{aligned} y_{t+1} &= y_t + \gamma(-x_t^{1/3} + U_t) \\ x_t &= \sup_{x \in [0,1]} \left( y_t x + 2\sqrt{x} \right) \end{aligned} \tag{C.72}$$

Solving the maximization problem in the definition of $x_t$, we obtain:

$$x_t = \begin{cases} (-y_t)^{-2}, & \text{if } y_t \le -1 \\ 1, & \text{if } y_t > -1 \end{cases} \tag{C.73}$$

or, equivalently, $x_t = \mathbb{1}(y_t > -1) + \mathbb{1}(y_t \le -1)(-y_t)^{-2}$ and the dual process can be written as:

$$y_{t+1} = y_t - \gamma \, \mathbb{1}(y_t > -1) - \gamma \, \mathbb{1}(y_t \le -1)(-y_t)^{-2/3} + \gamma U_t \tag{C.74}$$

For notational convenience, set $z_n \equiv -y_t$. Then, the evolution of the dual process becomes:

$$z_{t+1} = z_t + \gamma \, \mathbb{1}(z_t < 1) + \gamma \, \mathbb{1}(z_t \ge 1) z_t^{-2/3} - \gamma U_t \tag{C.75}$$

It is clear that $x_t \to 0$ if and only if $z_t \to \infty$ as $t$ goes to infinity. Now, define the process

$$z'_{t+1} = \left( z'_t + \gamma \, \mathbb{1}(z'_t < 1) + \gamma \, \mathbb{1}(z'_t \ge 1) z_t'^{-2/3} - \gamma U_t \right)^+, \quad z'_1 = z_1 \tag{C.76}$$

where $U_t$ is the same randomness as in (C.75).

**Claim 2.** *The process* $(z'_t)_{t \in \mathbb{N}}$ *dominates* $(z_t)_{t \in \mathbb{N}}$, *i.e.,* $z'_t \geq z_t$ *for all* $t \in \mathbb{N}$.

The proof of Claim 2 lies at the end. Now, invoking Theorem A.3 with

$$f(z) \equiv \gamma \mathbb{1}(z < 1) + \gamma \mathbb{1}(z \geq 1)z^{-2/3} \tag{C.77}$$

bounded, $\sigma^2 = \gamma^2$, and $\theta > 1$, we have

$$f(z) \geq \frac{\sigma^2 \theta}{2z} \quad \text{for all } z \text{ large enough} \tag{C.78}$$

Thus, $(z'_t)_{t \in \mathbb{N}}$ is transient, which implies that $\mathbb{P}(A) = 1$ for $A = \{\omega \in \Omega : \lim_{t \to \infty} z'_t(\omega) = \infty\}$.

Now, fix some $\omega \in A$. Since $\lim_{t \to \infty} z'_t(\omega) = \infty$, there exists $n_\omega \in \mathbb{N}$ such that $z'_t > 1$ for all $n \geq n_\omega$, and therefore

$$z'_{t+1} = z'_t + \gamma z'^{-2/3}_t - \gamma U_t$$

$$= z'_{n_\omega} + \gamma \sum_{s=n_\omega+1}^{t} \left( z'^{-2/3}_s - U_s \right) \tag{C.79}$$

from which we conclude that

$$\sum_{s=n_\omega+1}^{t} \left( z'^{-2/3}_s - U_s \right) \to \infty \quad \text{as } t \to \infty \tag{C.80}$$

Finally, we have that

$$z_{t+1} = z_{n_\omega} + \gamma \sum_{s=n_\omega+1}^{t} \left( \mathbb{1}(z_s < 1) + \mathbb{1}(z_s \geq 1)z^{-2/3}_s - U_s \right) \tag{C.81}$$

and, since $(z'_t)_{t \in \mathbb{N}}$ dominates $(z_t)_{t \in \mathbb{N}}$, and $z'_t > 1$ for all $t \geq t_\omega$, we readily get that

$$\sum_{s=n_\omega+1}^{t} \left( \mathbb{1}(z_s < 1) + \mathbb{1}(z_s \geq 1)z^{-2/3}_s - U_s \right) \geq \sum_{s=n_\omega+1}^{t} \left( z'^{-2/3}_s - U_s \right) \tag{C.82}$$

Thus, by (C.80), we conclude that

$$\sum_{s=n_\omega+1}^{t} \left( \mathbb{1}(z_s < 1) + \mathbb{1}(z_s \geq 1)z^{-2/3}_s - U_s \right) \to \infty \quad \text{as } t \to \infty \tag{C.83}$$

which implies that $\lim_{t \to \infty} z_t(\omega) = \infty$. Therefore, we obtain that $\lim_{t \to \infty} z_t(\omega) = \infty$ for all $\omega \in A$, and since $\mathbb{P}(A) = 1$, it follows that

$$\mathbb{P}\left( \lim_{t \to \infty} x_t = x^* \right) = \mathbb{P}\left( \lim_{t \to \infty} z_t = \infty \right) = 1 \tag{C.84}$$

and the proof is complete.

*Proof of Claim 2.* Consider the function

$$g(z) := z + \gamma \mathbb{1}(z < 1) + \gamma \mathbb{1}(z \geq 1)z^{-2/3} \tag{C.85}$$

Then:

- For $z < 1$: $g'(z) = 1 + \gamma > 0$

- For $z > 1$: $g'(z) = 1 - 2\gamma z^{-5/3}/3 > 1 - 2\gamma/3 > 0$

Thus, $g$ is strictly increasing for all $z \in \mathbb{R}$. Now, for the sake of contradiction, suppose that there exists $\omega \in \Omega$ and a first time $k + 1 \in \mathbb{N}$ where the dominance does not hold, i.e.,

$$z_k(\omega) \leq z'_k(\omega) \quad \text{and} \quad z'_{k+1}(\omega) < z_{k+1}(\omega) \tag{C.86}$$

By the monotonicity property of $g$, we get that

$$g(z_k(\omega)) \leq g(z'_k(\omega)) \tag{C.87}$$

and therefore, adding $-\gamma U_s(\omega)$ in both sides

$$z_{s+1}(\omega) \leq z'_s + \gamma \mathbb{1}(z'_s < 1) + \gamma \mathbb{1}(z'_s \geq 1)z'^{-2/3}_s - \gamma U_s$$

$$\leq \left( z'_s + \gamma \mathbb{1}(z'_s < 1) + \gamma \mathbb{1}(z'_s \geq 1)z'^{-2/3}_s - \gamma U_s \right)^+$$

$$\leq z'_{s+1}(\omega) \tag{C.88}$$

which is a contradiction. Thus, the proof of Claim 2 is complete. ∎

