# OpenReview forum: "Robust Equilibria in Continuous Games: From Strategic to Dynamic Robustness"
_NeurIPS.cc/2025/Conference — NeurIPS 2025 poster_

### Official Review · Reviewer_ezQj · 2025-06-06

**Clarity:** 4
**Significance:** 3
**Originality:** 3
**Rating:** 5
**Confidence:** 3

**Summary:**

This paper focuses on continuous games which exhibit some level of uncertainty and the stability of equilibria in these games. The authors provide the concept of a robust equilibrium to define and describe equilibria that remain valid under sufficiently small perturbations of the payoff structure. The paper provides several theoretical results, such as a geometric characterization of robust equilibria and the relation of strategic and dynamic robustness with respect to FTRL dynamics.

**Questions:**

1. Would it be possible to include an actual simulation of the examples?
2. How could the theoretical findings be leveraged to advance practical learning applications?

**Ethical Concerns:**

["NO or VERY MINOR ethics concerns only"]

**Final Justification:**

The authors did address my remaining questions adequately. Since my initial review and the evaluation of the other reviewers is mostly positive, I remain confident that the paper should be accepted for publication.

**Limitations:**

yes

**Paper Formatting Concerns:**

no formatting issues

**Quality:**

3

**Strengths And Weaknesses:**

Overall, I enjoyed reading the paper and did not spot any major weaknesses. Here are the details:

Strengths:
- The paper has a clear structure and extensively discusses related work which enables readers to access the contributions of the present work
- The authors consistently discuss limitations and provide context for their theoretical results
- The results seem to close a gap in the existing literature
- The theoretical results appear rigorous and assumptions are clearly stated
- Examples complement the theory

Suggestions:
- Would it be possible to include an actual simulation of the examples? I understand that the character of this paper is theoretical, but an exemplary computational simulation could provide further context
- I think there is a typo between lines 139 and 140: "for all $x_i$" instead of "for all $x$"

---

> ### Author Rebuttal · Authors · 2025-07-30
>
> Dear reviewer,
>
> Thank you for your time, constructive input and strong positive evaluation! We reply to your remarks and questions below:
>
> ---
> > Would it be possible to include an actual simulation of the examples? I understand that the character of this paper is theoretical, but an exemplary computational simulation could provide further context.
>
> The way that OpenReview has been set up this year, it is not possible to include any simulations during the rebuttal stage (either through a link to an external image or via an anonymized github / Jupyter notebook). However, we will be happy to present a simulation campaign examining the convergence of FTRL in atomic, non-splittable routing games with data taken from the well-known "TransportationNetworks" dataset of bstabler, which is a standard dataset for routing recommendation problems (we do not provide an external link to avoid violating the rebuttal guidelines).
>
> ---
> > How could the theoretical findings be leveraged to advance practical learning applications?
>
> Some practical learning applications where the theoretical findings can be leveraged are
> 1. **Markov potential games** in the context of multi-agent reinforcement learning.
> 2. **Atomic, non-splittable routing games** with arbitrary (increasing) cost functions.
> 3. **Cournot competitions** for agents with no (or small) price-setting power; etc.
> The robustness property implies that algorithms like FTRL can enjoy significantly faster convergence in such settings. However, our paper goes beyond simply identifying sufficient conditions for the convergence of FTRL. It also establishes that certain non-robust equilibrium configurations are ruled out as valid asymptotic outcomes of regularized learning. This insight highlights a broader practical concern: it helps explain why some learning problems, such as training GANs, remain inherently unstable and challenging.
>
> ---
> #### $^1$ *Justification of the existence of robust equilibria in the classes mentioned above:*
>
> 1. *For Markov potential games:* Theorem 3.1 of Leonardos et al. (2022) guarantees the existence of a determinstic Nash policy. This is an extreme point of the players' policy polytope, which is thus generically sharp (in the sense of Polyak, 1987), and hence robust.
> 2. *For atomic, non-splittable routing games:* see Rosenthal (1973) and note that strict Nash equilibria of finite games are robust.
> 3. *For Cournot competitions:* Nash-Cournot equilibria in Cournot games are characterized as solutions to a certain quadratic problem, with the quadratic coefficient determined by the agents' aggregate price-setting power (Monderer & Shapley, 1996). If the agents do not have price-setting power (or their price-setting power is relatively small), the solutions of said quadratic problem that do not exceed the agents' capacity are, by necessity, extreme, and hence generically robust.
>
> ---
> ### References
> - S. Leonardos, W. Overman, I. Panageas, and G. Piliouras, "Global convergence of multi-agent policy gradient in Markov potential games," ICLR 2022.
> - D. Monderer and L. S. Shapley, “*Potential games*,” Games and Economic Behavior, 14(1):124 – 143, 1996.
> - B. T. Polyak, *Introduction to Optimization.* New York, NY, USA: Optimization Software, 1987.
> - R. W. Rosenthal, “*A class of games possessing pure-strategy Nash equilibria*,” International Journal of Game Theory, vol. 2, pp. 65–67, 1973.
>
> > I think there is a typo between lines 139 and 140: "for all $x_i$" instead of "for all $x$"
>
> You are right, thanks for the catch!
>
> ---
> Thank you again for your time, input and positive evaluation—please do not hesitate to reach out if you have any further questions or remarks, we are looking forward to a constructive exchange during the discussion phase!
>
> Kind regards,
>
> The authors

---

> > ### Comment · Reviewer_ezQj · 2025-08-02
> >
> > I thank the authors for their informative and detailed responses! All of my questions have been answered and I will therefore keep my positive initial evaluation of the paper.

---

### Official Review · Reviewer_HTpi · 2025-06-13

**Clarity:** 4
**Significance:** 2
**Originality:** 2
**Rating:** 4
**Confidence:** 4

**Summary:**

This paper looks at the dynamics of the Follow The Regularized Leader (FTRL) algorithm in continuous games of finite dimension, in both deterministic and stochastic settings. It relies on an assumption of a robust equilibrium in order to ensure convergence under any regularizer. Especially, the rate is characterized depending on the regularizer, and FTRL is shown not to converge to non-equilibrium. The paper is purely theoretical.

**Questions:**

Could the assumptions on being a non-equilibrium for the non-convergence in Proposition 1 be relaxed? Proposition 3 shows that being non-robust is not enough, but perhaps a more precise characterization depending on the regularizer is possible? This could justify the very restrictive assumption of robustness.

**Ethical Concerns:**

["NO or VERY MINOR ethics concerns only"]

**Final Justification:**

While I still think the assumption of robustness is quite strong, the authors argued it is a common occurence in many games, and I may have misunderstood some techinicalities of the paper that made results appear simpler than they are.

**Limitations:**

Yes

**Paper Formatting Concerns:**

There is no formatting concern.

**Quality:**

3

**Strengths And Weaknesses:**

Strengths:

- The paper is well written and very clear.

- The statements are precise and rigorous. All properties are proven properly.

Weakness:

- The robust equilibrium assumption is very strong: it restricts the solutions to extreme points with gradients in the interior of the polar cone. As the main theorems assume that the primal iterates stay in a neighborhood of this extreme point, the results are relatively straightforward: in a neighborhood, all gradients are oriented "toward" this extreme point, and the trajectory of the dual iterates is easily characterized.

To support the idea of the restrictiveness of the assumption, in the specific case of mixed policies of a finite games, robustness is (if I'm not mistaken) equivalent to being a strict pure Nash equilibrium.

---

> ### Author Rebuttal · Authors · 2025-07-30
>
> Dear reviewer,
>
> Thank you for your time and input. We reply to your remarks and questions below:
>
> > The robust equilibrium assumption is very strong: it restricts the solutions to extreme points with gradients in the interior of the polar cone. As the main theorems assume that the primal iterates stay in a neighborhood of this extreme point, the results are relatively straightforward: in a neighborhood, all gradients are oriented "toward" this extreme point, and the trajectory of the dual iterates is easily characterized.
>
> We would like to clarify two points regarding this comment.
>
> First, in Theorem 2, **we do not assume** that "*the primal iterates stay in a neighborhood of this extreme point*". We only assume that FTRL is not *initalized* too far from said point, and the first important component of the proof is devoted precisely to establishing the stability of FTRL, i.e., that, with high probability, the induced sequence of play does not stray too far from equilibrium. [The key technical elements of this proof is Lemma C.1, which is in turn used to establish the stability bounds of Eqs. (C.25) and (C.26) in the proof of Theorem 2 in Appendix C.]
>
> By virtue of Theorem 2, the event $\mathcal{E} = \{\lim_{n\to\infty} x_n = x^\ast\}$ has probability $\geq 1-\delta$. For generality and notational convenience, Theorems 3 and 4 are predicated on this event but, again, the fact that the primal iterates of FTRL stay in a neighborhood of $x^\ast$ with high probability is not an assumption, *but something which we rigorously show* (and, in fact, one of the main technical challenges that we had to resolve).
>
> Second, the fact that gradients are oriented "toward" an extreme point does not suffice to establish convergence in a straightforward manner—in fact, this is false under different algorithms/setups.
>
> To see this, consider the simple, single-agent problem of maximizing the function $u(x) = x$ over $\mathcal{X} = [0,1]$ with the projected stochastic gradient ascent algorithm
> $$
> x_{n+1} = \Pi(x_n + \gamma \hat v_n)
> \qquad
> \textrm{(SGA)}
> $$
> where $\hat v_n$ is a stochastic gradient of $u$ at $x_n$, i.e., $\hat v_n = 1 + U_n$ for some noise process $U_n$. For simplicity, assume further that $U_n$ is a Bernoulli process with $U_n = \pm 1$ with probability $1/2$.
>
> In this toy example, $x^\ast = 1$ is a robust equilibrium. However, even if $x_n = x^\ast$ for some $n$, we would still have $x_{n+1} = 1-\gamma$ with probability $1/2$, so, by a straightforward application of the Borel-Cantelli lemma, we conclude that, with probability $1$, (SGA) does not converge to $x^\ast$.
>
> This shows that it is not straightforward to show that a given learning algorithm converges to a robust equilibrium in the presence of noise and uncertainty—and, in fact, outside the class of FTRL, this intuition is not correct.
>
> Finally, we should point out that many well-studied classes of games admit robust equilibria, for example:
> 1. **Atomic, non-splittable routing games** with arbitrary (increasing) cost functions
> 2. **Markov potential games** in the context of multi-agent reinforcement learning
> 3. **Cournot competitions** for agents with no (or small) price-setting power.
>
> We did not discuss these examples at length in the submitted version, but we will be happy to take advantage of the extra page and include a version of this discussion in the first revision opportunity.$^1$
>
> ---
> #### $^1$ *Justification of the existence of robust equilibria in the classes mentioned above:*
>
> 1. *For atomic, non-splittable routing games:* see Rosenthal (1973) and note that strict Nash equilibria of finite games are robust.
> 2. *For Markov potential games:* Theorem 3.1 of Leonardos et al. (2022) guarantees the existence of a determinstic Nash policy. This is an extreme point of the players' policy polytope, which is thus generically sharp (in the sense of Polyak, 1987), and hence robust.
> 3. *For Cournot competitions:* Nash-Cournot equilibria in Cournot competitions are characterized as solutions to a certain quadratic problem, with the quadratic coefficient determined by the agents' aggregate price-setting power (Monderer & Shapley, 1996). If the agents do not have price-setting power (or their price-setting power is small), the solutions of said quadratic problem that do not exceed the agents' capacity are, by necessity, extreme, and hence generically robust.
>
> ---
> > Could the assumptions on being a non-equilibrium for the non-convergence in Proposition 1 be relaxed? Proposition 3 shows that being non-robust is not enough, but perhaps a more precise characterization depending on the regularizer is possible? This could justify the very restrictive assumption of robustness.
>
> In the stated context, robustness is much more than an assumption / desideraturm, it is essentially a *necessary* condition. Specifically, Proposition 2 shows that if the noise affecting FTRL is persistent (in that it has positive covariance throughout), *the sequence of play generated by FRTL cannot converge with positive probability to an interior equilibrium*—or, more generally, to a non-extreme equilibrium.
>
> Together with Proposition 3, these results imply that only robust equilibria can admit robust convergence guarantees. The proof of Proposition 3 required introducing results from Markov chain theory on the recurrence versus transience of a biased random walk with vanishing bias—tools that, to the best of our knowledge, have not been applied in this context before. This result also highlights the technical difficulty of the problem: even for a simple one-player, one-dimensional examples, establishing non-convergence requires substantial machinery and novel tecnhiques. A complete characterization of convergence behavior based on the choice of regularizer remains an open and highly challenging problem, and our work provides a suggestive first step in that direction.
>
> ---
> Thank you again for your time and input—we are looking forward to further constructive exchanges during the rebuttal phase.
>
> Kind regards,
>
> The authors
>
> ---
> ### References
> - S. Leonardos, W. Overman, I. Panageas, and G. Piliouras, "Global convergence of multi-agent policy gradient in Markov potential games," ICLR 2022.
> - D. Monderer and L. S. Shapley, “*Potential games*,” Games and Economic Behavior, 14(1):124 – 143, 1996.
> - B. T. Polyak, *Introduction to Optimization.* New York, NY, USA: Optimization Software, 1987.
> - R. W. Rosenthal, “*A class of games possessing pure-strategy Nash equilibria*,” International Journal of Game Theory, vol. 2, pp. 65–67, 1973.

---

> > ### Comment · Reviewer_HTpi · 2025-08-04
> >
> > Thank you for your detailed response. I may have misunderstood some key points of the paper, especially in Theorem 2.
> >
> > I updated my score accordingly.

---

> > > ### Author Response · Authors · 2025-08-06
> > >
> > > Our sincere thanks for your time, your thoughtful input and your comments, we are glad we addressed your questions.
> > >
> > > Warm regards,
> > >
> > > The authors

---

### Official Review · Reviewer_5rLL · 2025-06-23

**Clarity:** 3
**Significance:** 2
**Originality:** 3
**Rating:** 5
**Confidence:** 3

**Summary:**

This paper defines two notions of stability for local Nash equilibrium in continuous games: strategic robustness and dynamic robustness. For strategic robustness, the author refers to a Nash equilibrium that remains unchanged under perturbation in the underlying game's payoff structure in terms of the vector fields. For dynamic robustness, the authors refer to an equilibrium that attracts the FTRL dynamics with initial conditions in a neighborhood.

The main results include a characterization of the strategic robustness equilibrium (Thm 1) and a proof that FTRL converges to the strategic robustness equilibrium (Thm 2).

**Questions:**

* Can the author provide a characterization of the existence of a robust Nash equilibrium as defined in Definition 1? For example, in which kind of important and interesting games does the robust Nash equilibrium necessarily exist?

  **(This is my main concern. I would be happy to increase my evaluation score if this issue is addressed.)**

* Does the dynamic robustness result still hold when considering the average-iterate behavior of FTRL dynamics?

* Is it possible to define robustness for mix Nash equilibrium?

**Ethical Concerns:**

["NO or VERY MINOR ethics concerns only"]

**Final Justification:**

My main concern regarding the existence of a robust equilibrium defined in this paper has been addressed.

**Limitations:**

yes

**Quality:**

3

**Strengths And Weaknesses:**

Strengths:

* The problem studied in this paper is interesting, and the writting is clear and easy to follow.
* The observation that small perturbation on the payoff value of games will destroy all the equilibrium (Example 3.1 & 3.2) is interesting.
* Theorem 1 provides an interesting characterization of the geometry of robust equilibrium in terms of TC and PC.

Weaknesses:

* It is not clear which kinds of games could admit a robust equilibrium. Given that the existence of a robust equilibrium is a precondition for the convergence result in Thm 2, the lack of discussion on the existence of a robust equilibrium makes the value of this result less clear. For example, it seems that all the interior equilibria will not be robust according to Thm 1, which implies that a robust equilibrium may not exist in many interesting unconstrained games, such as GANs.

* It is not very well-motivated to consider the constant step size FTRL dynamics, given that the vanishing step size is a benchmark in the theoretical analysis of the convergence properties of FTRL. Especially, it is not clear whether the result of Theorem 2 also applies to the vanishing step size case.

---

> ### Author Rebuttal · Authors · 2025-07-30
>
> Dear reviewer,
>
> Thank you for your time and constructive input. We reply to your remarks and questions below:
>
> > It is not clear which kinds of games could admit a robust equilibrium. Given that the existence of a robust equilibrium is a precondition for the convergence result in Thm 2, the lack of discussion on the existence of a robust equilibrium makes the value of this result less clear.
> [...]
> > Can the author provide a characterization of the existence of a robust Nash equilibrium as defined in Definition 1? For example, in which kind of important and interesting games does the robust Nash equilibrium necessarily exist? (This is my main concern. I would be happy to increase my evaluation score if this issue is addressed.)
>
> Certainly! Some widely studied classes of games that always admit robust equilibria (except for a measure zero set of instances) are as follows:
> 1. **Atomic, non-splittable routing games** with arbitrary (increasing) cost functions.
> 2. **Markov potential games** in the context of multi-agent reinforcement learning.
> 3. **Cournot competitions** for agents with no (or small) price-setting power.
>
> We did not discuss these examples at length in the submitted version, but we will be happy to take advantage of the extra page and include a version of this discussion in the first revision opportunity.$^1$
>
> ---
> #### $^1$ *Justification of the existence of robust equilibria in the classes mentioned above:*
>
> 1. *For atomic, non-splittable routing games:* see Rosenthal (1973) and note that strict Nash equilibria of finite games are robust.
> 1. *For Markov potential games:* Theorem 3.1 of Leonardos et al. (2022) guarantees the existence of a determinstic Nash policy. This is an extreme point of the players' policy polytope, which is thus generically sharp (in the sense of Polyak, 1987), and hence robust (by the characterization of Theorem 1).
> 1. *For Cournot oligopolies:* Nash-Cournot equilibria in Cournot games are characterized as solutions to a certain quadratic problem, with the quadratic coefficient determined by the agents' aggregate price-setting power (Monderer & Shapley, 1996). If the agents do not have price-setting power (or their price-setting power is small), the solutions of said quadratic problem that do not exceed the agents' capacity are, by necessity, extreme, so the resulting equilibria are generically robust.
>
> ---
> > For example, it seems that all the interior equilibria will not be robust according to Thm 1, which implies that a robust equilibrium may not exist in many interesting unconstrained games, such as GANs.
>
> Yes, this is precisely the point! Proposition 2 shows that, under persistent noise, **the sequence of play generated by FTRL with a non-vanishing step-size cannot converge with positive probability to an interior equilibrium**—or, more generally, to non-extreme equilibria.
>
> Put differently, our paper does not simply provide a sufficient condition for convergence of FTRL, but it also rules out other, non-robust equilibrium configurations as valid limits of regularized learning. This is a cautionary tale which we believe carries important insights for practitioners because it provides a plausible explanation as to why certain machine learning models (including GANs) are notoriously difficult to train.
>
> ---
> > It is not very well-motivated to consider the constant step size FTRL dynamics, given that the vanishing step size is a benchmark in the theoretical analysis of the convergence properties of FTRL.
>
> We should first point out that our results have been stated with a constant step-size only for clarity and simplicity of presentation. In particular, **Theorems 2-4 apply verbatim to FTRL with a variable, *non-constant* step-size schedule $\gamma_n$ with $\gamma:= \sup_n \gamma_n$ and $\liminf_n \gamma_n >0$.**
>
> [The required modifications in the proofs are trivial, but we would be happy to detail them if you want, and we will of course be happy to adapt our presentation in the first revision opportunity]
>
> Now, the use of a vanishing step-size schedule—typically satisfying a Robbins-Monro summability condition like $\sum_n \gamma_n^2 < \sum_n \gamma_n =\infty$ or a variant thereof—is indeed common in the stochastic approximation literature. At the same time however, it introduces an inverse recency bias, in that **new information enters the algorithm with decreasing weights**. This feature of vanishing step-size schedule contradicts "*the general principles of iterative schemes, in accordance to which new information is more important than old one*" (Nesterov, 2009, p. 224), and is well-known to suffer from important drawbacks, both theoretical and practical:
>
> - From a theory standpoint, it is known that FTRL with a vanishing step-size schedule may lead to **superlinear regret**, even in simple instances like the exponential weights algorithm (Orabona and Pál, 2018). This is an important part of the reason that FTRL was introduced and analyzed in online learning with a *constant* step-size, cf. the textbooks of Shalev-Shwartz (2011) and Lattimore & Szepesvári (2020).
> - From a practical standpoint, gradient methods with a vanishing step-size suffer from slow warm-up periods, and take a long time to converge to a neighborhood of an equilibrium point. By contrast, in machine learning setups, constant step-size methods reach the vicinity of a solution very fast (often within  0.1% accuracy), cf. Dieuleveut et al. (2020).
>
> For this reason, in many state-of-the-art transformers and LLMs, step-size schedules are often constant over billions (or even trillions) of samples. For example, DeepSeek is trained as follows (DeepSeek v3 technical report, Section 4.2):
>
> ```
> We keep a constant learning rate of 2.2 × 10−4 until the model consumes 10T training tokens.
> [...]
> During the training of the final 500B tokens, we keep a constant learning rate of 2.2 × 10−5 in the first 333B tokens, and switch to another constant learning rate of 7.3 × 10−6 in the remaining 167B tokens.
> ```
>
> Of course, there are no silver bullets in optimization: we do not claim that non-vanishing step-size schedules are universally superior to vanishing ones (each has its trade-offs, advantages and disadvantages) but, overall, the use of non-vanishing schedules is very strongly grounded in both theory and practice.
>
> ---
> > It is not clear whether the result of Theorem 2 also applies to the vanishing step size case.
>
> Yes, it actually does! The key observation is that Lemma C.1 continues to hold under a vanishing step-size rule with the RHS of (C.8) replaced by
> $$
> \frac{\sum_{k=1}^n \gamma_k^{q(1-\mu) \sigma_k^q}}{(\sum_{k=1}^n \gamma_k)^{1+q(\mu-1/2)}}
> $$
> If coupled with a Robbins-Monro / Kiefer-Wolfowitz summability condition, this modified version of Lemma C.1 is sufficient for the proof of Theorem 2 to go through essentially verbatim, replacing $\gamma n$ with $\sum_{k=1}^n \gamma_k$.
>
> We did not tackle this case because we wanted to keep the analysis as simple as possible, but we will be happy to include the above.
>
> ---
> > Does the dynamic robustness result still hold when considering the average-iterate behavior of FTRL dynamics?
>
> Theorems 2-4 hold automatically for the time-average of $x_n$ (since $x_n$ converges, so does its time-average). However, the rate for the time-average would be much slower than (16), since time-averages cannot converge at a rate faster than $\mathcal{O}(1/t)$.
>
> ---
> > Is it possible to define robustness for mix Nash equilibrium?
>
> In the context of finite games, mixed Nash equilibria are interior, so they are not robust.
>
>
> ---
> Thank you again for your time and input—we look forward to further constructive exchanges during the discussion phase.
>
> Kind regards,
>
> The authors
>
> ---
> ### References
> - A. Dieuleveut, A. Durmus, and F. Bach, "*Bridging the gap between constant step size stochastic gradient descent and Markov chains*," The Annals of Statistics, vol. 48, no. 3, 2020.
> - T. Lattimore and C. Szepesvári, *Bandit Algorithms*. Cambridge, UK: Cambridge University Press, 2020.
> - S. Leonardos, W. Overman, I. Panageas, and G. Piliouras, "Global convergence of multi-agent policy gradient in Markov potential games," ICLR 2022.
> - D. Monderer and L. S. Shapley, “*Potential games*,” Games and Economic Behavior, 14(1):124 – 143, 1996.
> - Y. Nesterov, “Primal-dual subgradient methods for convex problems,” Mathematical Programming, vol. 120, no. 1, pp. 221–259, 2009.
> - F. Orabona and D. Pál, “*Scale-free online learning*,” Theoretical Computer Science, vol. 716, pp. 50–69, 2018.
> - B. T. Polyak, *Introduction to Optimization*. New York, NY, USA: Optimization Software, 1987.
> - R. W. Rosenthal, “*A class of games possessing pure-strategy Nash equilibria*,” International Journal of Game Theory, vol. 2, pp. 65–67, 1973.

---

> > ### Comment · Reviewer_5rLL · 2025-08-02
> >
> > I would like to sincerely thank the authors for their thorough and thoughtful rebuttal. My primary concern regarding the existence of a robust equilibrium has been adequately addressed, which has led me to raise my score to 5. I highly encourage the authors to incorporate the valuable content from their rebuttal into the next version of the paper.

---

### Official Review · Reviewer_D3Vd · 2025-07-03

**Clarity:** 3
**Significance:** 4
**Originality:** 4
**Rating:** 6
**Confidence:** 3

**Summary:**

The paper shows connections between two notions of robustness in games. The first is strategic robustness which requires that an equilibrium of a game continues to be an equilibrium even if the payoffs of the game are perturbed by a small amount. The second is dynamic robustness, which requires that learning dynamics converge to this equilibrium even if the initial point is perturbed by a small amount.

The second notion of robustness is interesting in dynamics where the step-sizes don’t decay. So the paper considers dynamic robustness with FTRL as learning dynamics and show that their results apply to FTRL with any choice of regularizer.

The paper provides a geometric interpretation of strategic robust equilibria. The paper shows that strategic robustness is a sufficient condition for dynamic robustness relative to FTRL for two feedback structures: unbiased estimate of loss gradients and simply the loss received.

Toward arguing that strategic robustness is also necessary, they show that strategic robustness is necessary for some regularizer in an example they construct.

**Questions:**

Do you think this connection will extend to other dynamics, such as gradient descent?

**Ethical Concerns:**

["NO or VERY MINOR ethics concerns only"]

**Final Justification:**

The work contained interesting results and was presented well. They also provided helpful clarifications and addressed concerns during the rebuttal phase. Hence I would like to keep my positive score.

**Limitations:**

Yes

**Quality:**

4

**Strengths And Weaknesses:**

Strengths: The paper makes a nice connection between two forms of robustness of equilibria. This also seems to be a connection between two forms of equilibrium selection --- based on payoffs and based on convergence of dynamics. The paper is well-written and the main arguments are easy to follow.

Weaknesses: There could be more intuition provided about the proof for Theorem 2 which is the main theorem showing sufficiency of strategic robustness for dynamic robustness. Namely, describing the main technical ideas/tools to analyze convergence of FTRL and how strategic robustness enters this would be helpful.

---

> ### Author Rebuttal · Authors · 2025-07-30
>
> Dear reviewer,
>
> Thank you for your time, detailed input, and strong positive evaluation! We reply to your remarks and questions below:
>
> ---
> > There could be more intuition provided about the proof for Theorem 2 which is the main theorem showing sufficiency of strategic robustness for dynamic robustness. Namely, describing the main technical ideas/tools to analyze convergence of FTRL and how strategic robustness enters this would be helpful.
>
> Sure thing. The key idea is that, under the conditions of Theorem 2, the auxiliary process $y_n$ (which aggregates the players' individual gradient steps) escapes to infinity along a direction which is "just right" in the sense that it carries the sequence of chosen actions $x_n = Q(y_n)$ toward the equilibrium in qeustion.
>
> A bit more formally, the intuition and key steps for the proof are as follows:
>
> 1. From Theorem 1 (and the continuity of the players' payoff functions), strategic robustness of an equilibrium $x^\ast$ implies that the players' individual gradient field points toward $x^\ast$ in a neighborhood thereof.
> 1. As a result, the "gradient update" component of the FTRL algorithm—that is, the $y_{n+1} = y_n + \gamma \hat v_n$ part of the process—aggregates gradient steps that are, on average, aligned along the interior of the normal cone $\mathrm{NC}(x^\ast)$ to the players' action space $\mathcal{X}$ at $x^\ast$.
> 1. Because of this alignment, the process $y_n$ exhibits a consistent drift that takes it deeper and deeper toward a "copy" of the normal cone $\mathrm{NC}(x^\ast)$ that lives in the gradient space $\mathcal{Y}$. [The consistency of this drift is not easy to show: this is the subject of Lemma C.1 in Appendix C, and the first part of the proof of Theorem 2, again in Appendix C]
> 1. In tandem to the above, we show that, with very high probability, the process never strays too far from $x^\ast$. [Again, this is quite technical: it requires a tandem application of Proposition C.1 / Corollary C.1, and the supremum estimation of Eqs. (C.25) and (C.26) in Appendix C.]
> 1. Combining all of the above, Proposition 1 allows us to show that the sequence of chosen actions $x_n = Q(y_n)$ under (FTRL) converges to $x^\ast$.
>
> Please let us know if this is clear enough! We will be happy to include a version of this discussion in the first revision opportunity—we did not do so in the submitted version of the paper due to lack of space.
>
> ---
> > Do you think this connection will extend to other dynamics, such as gradient descent?
>
> Great question—both yes and no.
>
> To explain what we mean, consider the unconstrained (stochastic) gradient ascent algorithm
> $$
> x_{n+1} = x_n + \gamma \hat v_n
> \qquad
> \text{(SGA)}
> $$
> where $\hat v_n$ is some stochastic payoff gradient. **The gradient algorithm (SGA) is captured directly from our setup in the case $\mathcal{X} = \mathbb{R}^d$ by taking $h(x) = \|x\|^2/2$** (cf. Example 2.2 in L178 our paper).
>
> Now, in the constrained case, there are two variants of (SGA) to consider. The first interleaves gradient steps and projections as
> $$
> x_{n+1} = \Pi(x_n + \gamma \hat v_n)
> \qquad
> \textrm{(P-SGA)}
> $$
> where $\Pi$ denotes the standard Euclidan projection on $\mathcal{X}$. The second is known as "SGA with lazy projections" (Zinkevich, 2003) and proceeds as
> $$
> y_{n+1} = y_n + \gamma \hat v_n,
> \quad
> x_{n+1} = \Pi(x_n)
> \qquad
> \textrm{(L-SGA)}
> $$
> i.e., the algorithm aggregates gradient steps and *then* projects the result onto $\mathcal{X}$, cf. Zinkevich (2003), Nesterov (2009), Shalev-Shwartz (2011) and references therein.
>
> **The latter version of the stochastic gradient dynamics (L-SGA) is one of the archetypal examples of FTRL.** We mentioned this in Example 2.2 (L178), but there was no space to expand upon this in detail—we would be more than happy to include a detailed discussion along the above lines at the first revision opportunity.
>
> ---
> Thank you again for your constructive input, encouraging words and strong positive evaluation—please do not hesitate to reach out if you have any further questions!
>
> Kind regards,
>
> The authors
>
> ---
> ### References
>
> - Y. Nesterov, “*Primal-dual subgradient methods for convex problems*,” Mathematical Programming, vol. 120, no. 1, pp. 221–259, 2009.
> - S. Shalev-Shwartz, “*Online learning and online convex optimization*,” Foundations and Trends in Machine Learning, vol. 4, no. 2, pp. 107–194, 2011.
> - M. Zinkevich, “*Online convex programming and generalized infinitesimal gradient ascent*,” in ICML ’03: Proceedings of the 20th International Conference on Machine Learning, pp. 928–936, 2003.

---

### Decision · Program_Chairs · 2025-09-17

**Decision:**

Accept (poster)

**Comment:**

This paper studies the robustness of Nash equilibrium in continuous games, under both strategic and dynamic uncertainty. A robust Nash equilibrium of a game is the one that remains an equilibrium even the payoff functions of the game changes arbitrarily by a small amount. The paper carefully justifies the distance notion of closeness between two games when they quantify changes in the payoff functions. They provide a geometric interpretation of the robust equilibrium. They then establish the results for dynamic robustness -- aiming at answering the question: Which equilibria remains attracting the dynamics of FTRL under small perturbations to the initialization of the dynamic's initial points. They establish that any strategically robust equilibrium has the property of robust dynamics.

Strengths. A very well written paper with a good trace of history about the quest for defining meaningful equilibrium, making the studied problem significant, a fairly complete characterization of robustness and the relationship between strategic robustness and dynamic robustness.

Weaknesses. No major concerns from the reviews as they are well addressed by the authors. For me, the only concern is that, for dynamic robustness, it requires that the initial point is initialized within a neighborhood of the convergent point, which makes the dynamic robustness less appealing, as my first impression is that the perturbation to the initial point is within a neighborhood centered at any point, though technically I am not sure if that result is possible.

Reasons for accept. See Strengths

During the rebuttal and discussion, the paper received positive evaluations. This is evident by that two reviewers have increased their scores after the discussion.